# Hierarchical assembly of tryptophan zipper peptides into stress-relaxing bioactive hydrogels

Ashley K. Nguyen[1,2], Thomas G. Molley[1,2,3], Egi Kardia[4,5,6], Sylvia Ganda[1,2], Sudip Chakraborty[1], Sharon L. Wong[4,5,6], Juanfang Ruan [7], Bethany E. Yee[1,2], Jitendra Mata [1,8], Abhishek Vijayan[4,5,9], Naresh Kumar[1], Richard D. Tilley [1,7], Shafagh A. Waters[2,4,5,6,10] & Kristopher A. Kilian [1,2,3,6] ✉

Soft materials in nature are formed through reversible supramolecular assembly of biological polymers into dynamic hierarchical networks. Rational design has led to self-assembling peptides with structural similarities to natural materials. However, recreating the dynamic functional properties inherent to natural systems remains challenging. Here we report the discovery of a short peptide based on the tryptophan zipper (trpzip) motif, that shows multiscale hierarchical ordering that leads to emergent dynamic properties. Trpzip hydrogels are antimicrobial and self-healing, with tunable viscoelasticity and unique yield-stress properties that allow immediate harvest of embedded cells through a flick of the wrist. This characteristic makes Trpzip hydrogels amenable to syringe extrusion, which we demonstrate with examples of cell delivery and bioprinting. Trpzip hydrogels display innate bioactivity, allowing propagation of human intestinal organoids with apical-basal polarization. Considering these extensive attributes, we anticipate the Trpzip motif will prove a versatile building block for supramolecular assembly of soft materials for biotechnology and medicine.

Natural extracellular matrices are composed of a meshwork of multiple proteins with an interconnected and hierarchical structure that are ideal for guiding tissue assembly. However, the in vitro use of these matrices is hampered by batch-to-batch variability and concerns of immunogenicity[1,2]. Thus, the discovery of synthetic alternatives remains a principal goal for cell biologists and tissue engineers. In pursuit of this, hydrogels comprised of self-assembling synthetic peptides have attracted broad interest due to the uniformity of

starting material, ease of synthesis, biodegradability, and low cytotoxicity[3–5]. These synthetic hydrogelators have been designed to form entangled networks of peptide nanofibers that mimic the structural characteristics of native matrices, including mesh size, pore size, and nanofiber architecture[6,7].

New peptide hydrogelators are commonly designed via functionalization of ultra-short hydrophobic peptides (2–5 residues), with large N-terminal capping groups that favor self-assembly through pi-

[1]School of Chemistry, University of New South Wales, Sydney, NSW 2052, Australia. [2]Australian Center for Nanomedicine, University of New South Wales, Sydney, NSW 2052, Australia. [3]School of Materials Science and Engineering, University of New South Wales Sydney, Sydney, NSW 2052, Australia. [4]School of Biomedical Sciences, Faculty of Medicine and Health, University of New South Wales, Sydney, NSW 2052, Australia. [5]Molecular and Integrative Cystic Fibrosis Research Centre (miCF_RC), University of New South Wales, Sydney, NSW 2052, Australia. [6]School of Clinical Medicine, Faculty of Medicine and Health, University of New South Wales, Sydney, NSW 2052, Australia. [7]Electron Microscopy Unit, Mark Wainwright Analytical Centre, University of New South Wales, Sydney, NSW 2052, Australia. [8]Australian Centre for Neutron Scattering, Australian Nuclear Science and Technology Organization, Lucas Heights, NSW 2234, Australia. [9]School of Biotechnology and Biomolecular Sciences, University of New South Wales, Sydney, NSW 2052, Australia. [10]Department of Respiratory Medicine, Sydney Children's Hospital, Randwick, NSW 2031, Australia. ✉e-mail: k.kilian@unsw.edu.au

stacking interactions[8]. Peptide amphiphiles are another class of gelators, relying on an arrangement of hydrophobic and hydrophilic domains to drive self-assembly[9,10]. Self-complementary ionic gelators are typically formed from longer peptides (8–16 residues) with alternating charged amino acids that self-assemble through electrostatic interactions[11]. Alternative approaches involve rational design of peptide sequences that imitate naturally occurring secondary protein structures, such as the alpha helix[12] or beta sheet[13]. Despite extensive research into supramolecular peptide assembly, the discovery of new hydrogelators is most often driven by serendipity or permutation of pre-existing gelator sequences.

A relatively unexplored assembly motif is the tryptophan zipper (Trpzip). This motif is characterized by four cross-strand tryptophan residues that interlock via the indole rings, folding the peptide into a beta hairpin conformation[14]. As a result of this highly stabilizing zipper effect, beta hairpins can be formed from Trpzip peptides that are as short as twelve amino acids. This is considerably shorter than previously reported beta hairpin hydrogelators, such as MAX peptides, which rely on the tetrapeptide (-V$^{D}$PPT-) unit and lengthier sequences (>20 amino acids) to maintain a folded hairpin structure[15]. Previous work has shown the promise of tryptophan as a building block for self-assembled hydrogels; however, these systems still rely on the combined use of other non-natural hydrophobic moieties (e.g., benzene or naphthyl residues)[16,17] or conventional peptide gelator motifs (e.g., diphenylalanine) to help drive gelation[18]. Although Trpzip sequences have only been shown to assemble into nanofibers over the course of several weeks[19], its potential as a new self-assembling structural motif is evident. However, the Trpzip peptide motif has not yet been used to form hydrogels.

In this work, we utilized molecular dynamics simulations to design a peptide hydrogelator based on the Trpzip motif. Our Trpzip peptides undergo hierarchical assembly within minutes, first forming long nanofibers, followed by assembly into microscale domains with periodic architecture. The resulting hydrogel showed thermoresponsive gelation with tunable modulus, self-healing, and stress-relaxing characteristics, cell viability and spreading even without cell adhesion motifs, and antimicrobial activity. Adding short adhesion peptides have minimal effect on the mechanical properties of the hydrogel and adding extracellular matrix proteins such as laminin enabled culture of adult stem cell and induced pluripotent stem cell derived intestinal organoids. The low yield-strain of the material facilitates rapid fluidization under shear, providing a simple mechanism to retrieve embedded cells and tissue and to disperse cell-laden Trpzip gels via syringe towards cell delivery and bioprinting applications.

## Results and discussion

### Computational screening identifies self-assembling tryptophan zipper variants

The Trpzip peptide motif has enabled the synthesis of the shortest reported beta hairpins to date (Fig. 1a)[14]. Originally designed for studying the thermodynamics of protein folding, minor changes in Trpzip peptide sequences have been shown to drastically alter aggregation and nanofiber formation (Fig. 1b). Markiewicz et al. proposed that the positively charged lysine residue in the eighth position acted as an aggregation gatekeeper by preventing peptide monomer association through repulsive forces[19]. To test this hypothesis, we ran coarse-grain molecular dynamics (MD) simulations of Trpzip variants using all twenty canonical amino acids, with the prediction that small uncharged residues in place of lysine may be most likely to facilitate nanofiber formation.

MD simulations were run for all twenty Trpzip1 peptide variants, and the last frame of each simulation were visually assessed (Fig. 1c). Comparing all variants by extracting the largest peptide aggregate from each simulation and aligning them on the same axis, we found the valine substitution showed the largest difference between aggregation

modes compared to the original sequence (Fig. 1d), which we named Trpzip-V. Quantifying the moments of inertia of the largest peptide clusters for all twenty variants confirmed the valine-substituted variant formed large aggregates the most fibrous in morphology, while Trpzip1 formed small spherical aggregates (Fig. 1e). Backmapping to all-atom resolution similarly indicated aggregation of Trpzip-V monomers formed fibrillar nanostructures (Supplementary Fig. 1).

### Hydrogel assembly arises from multiscale hierarchical order

Next, we synthesized Trpzip-V to experimentally assess its potential for self-assembly under physiological conditions (Supplementary Fig. 2). Trpzip-V was able to form a gel under both acidic and basic pH conditions (Supplementary Fig. 3a) but precipitated out of solution at neutral pH. We attributed the loss in solubility at neutral pH to be due to the peptide's isoelectric point of 6.97. Therefore, a second variant was designed with an uncharged glutamine residue to replace the negatively charged glutamic acid (Trpzip-QV; Fig. 2a). This new variant has an overall charge of +1 at pH 7 to enable solubility at neutral pH (Supplementary Fig. 3b). After overnight incubation at 37 °C in pure water, Trpzip-QV (0.1% w/v) formed a self-supporting hydrogel while Trpzip1 remained liquid (Fig. 2a). Circular dichroism (CD) spectroscopy confirmed Trpzip-QV still folds into a beta hairpin, as evidenced by the positive band at 228 nm, which is indicative of tryptophan's indole rings packing into a hairpin conformation (Fig. 2b). Additionally, Trpzip-QV showed a higher level of aggregation after 24 h compared to Trpzip1, as revealed by a lower CD signal at 228 nm, which is correlated with the reorganization of the peptide hairpin into fibrillar aggregates[20,21]. For simplicity, Trpzip-QV will be referred to as Trpzip hereafter.

To understand the mechanism behind gelation, we performed transmission electron microscopy (TEM), which revealed an entangled network of fibrillar nanostructures, approximately $4.12 \pm 1.03$ nm in diameter (Fig. 2c; Supplementary Fig. 4). To visualize the nanostructures under hydrated conditions, Trpzip gels were imaged over 24 h using cryo-TEM, demonstrating gradual assembly and elongation of nanofibers (Fig. 2d; Supplementary Fig. 5). Additionally, Fourier transform infrared (FTIR) spectra showed prominent peaks in the Amide I (1600–1800 cm$^{-1}$) and Amide II (1500–1600 cm$^{-1}$) regions (Supplementary Fig. 6), suggesting gelation is driven by beta sheet aggregation of the Trpzip monomers, consistent with the aggregation modes for other beta hairpin gelators reported[22].

To further probe the self-assembly pathway of Trpzip hydrogels, we performed small angle neutron scattering (SANS). The data was fitted initially with the shape-independent power-law model, which revealed the slope in the high Q region was −2 (Fig. 2e; far left). This suggested the formation of lamellae or disc-shaped particles on the order of 1–10 nm. The slope in the low Q region was found to be −3.7, indicating the formation of large mass fractal-like aggregates exceeding 500 nm in size. These large fractal aggregates continue to increase in size over the course of seven days, as indicated by the decrease in slope in the low Q region. This was further confirmed by ultra-small angle neutron scattering (USANS) measurements (Supplementary Fig. 7). Data fitting to a lamellae model indicates the size of the lamellae particles is approximately 2.2 nm. At higher hydrogel concentrations (3% w/v), a decrease in intensity in the high Q region indicates the assembly of larger fractal aggregates (Fig. 2e; center left). The scattering profile of Trpzip gels fit well with a combination of the lamellae model in the high Q region and the power law fit in the low Q region (Fig. 2e; center right and far right), as further confirmation of the presence of both nanometer-scale lamellae particles and micrometer-scale fractal aggregates.

To explore the microscale assembly, we performed scanning electron microscopy (SEM). Gelation at basic pH produced a highly homogenous network of lamellae stacks, with fibrillar nanostructures on the order of a few nanometers comprising this meshwork (Fig. 2f;

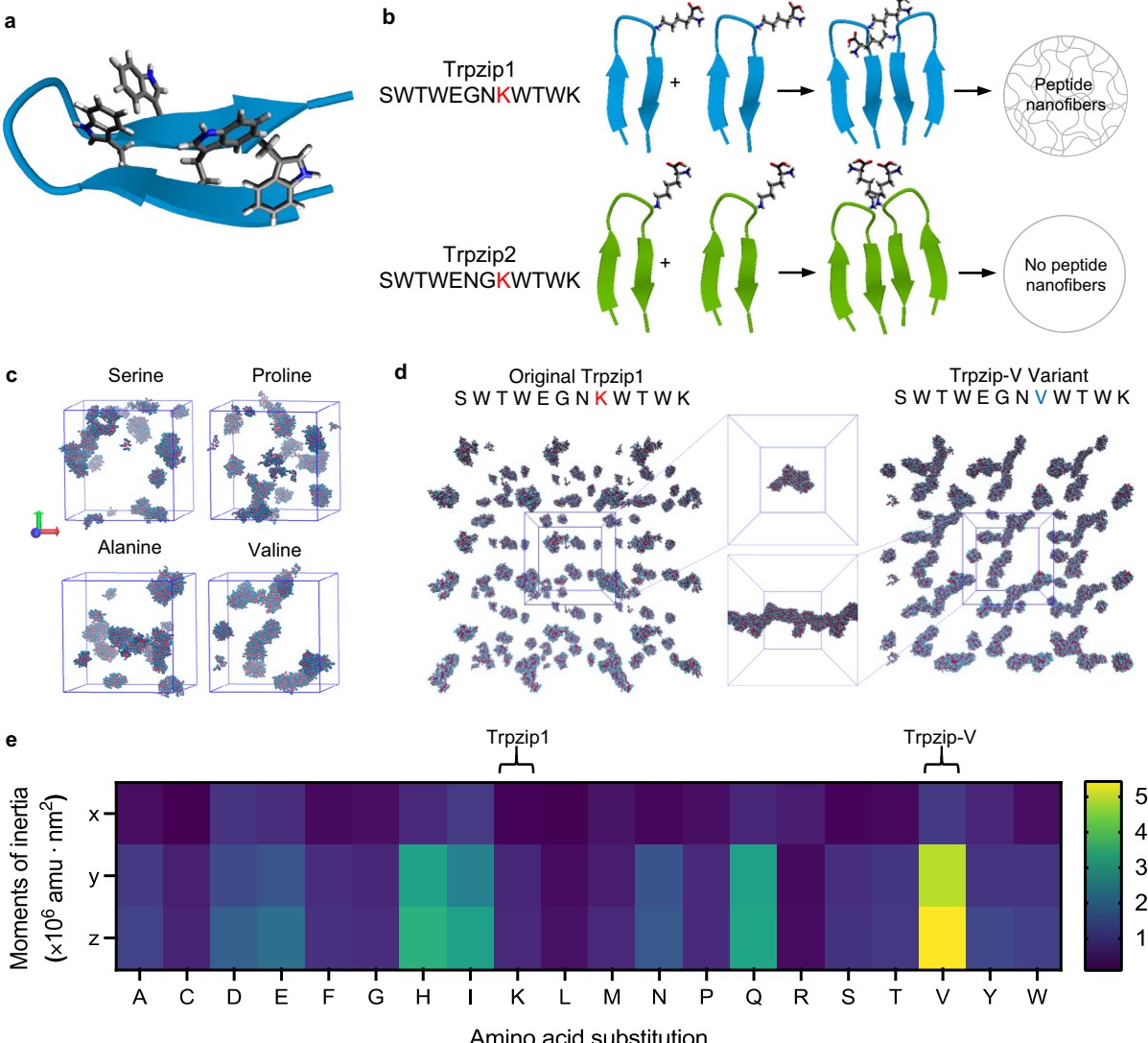

**Fig. 1 | Coarse-grain molecular dynamics simulations to identify Trpzip sequences prone to nanofiber aggregation. a** Cartoon representation of the tryptophan zipper folded in a beta hairpin conformation, with interlocking tryptophan residues depicted. **b** Schematic demonstrating the role of lysine as an aggregation gatekeeper in Trpzip1 and Trpzip2, and the effect of successful aggregation gatekeeping on the nanofiber-forming ability of the peptides. **c** Results of coarse-grain simulations run for Trpzip variants with serine, proline, alanine, or valine in the place of the eighth lysine residue present in the original Trpzip1 sequence. **d** Comparison of coarse-grain simulations run for the original Trpzip1 and Trpzip-V variant. Center boxes show the largest cluster of peptides extracted from each simulation and aligned on the same axis for visual comparison of size, length, and fibril morphology of peptide aggregates. **e** Moments of inertia were calculated for all variants generated using the twenty canonical amino acids.

Supplementary Fig. 8). Upon balancing to neutral pH, this highly organized, discrete honeycomb network is disrupted, and the resulting gel network is reformed through entanglement of more irregularly shaped nanofibers. Based on the combined findings, we propose Trpzip-QV peptides self-assemble into hydrogels via the following mechanism. Peptide monomers fold into a beta hairpin conformation and assemble into a disc/ellipsoidal-shaped particle, as suggested by SANS. These particles stack via peptide backbone interactions to create nanofibers, aligning to create a meshwork of lamellae stacks which give rise to the macro-scale order (Fig. 2g).

**Trpzip hydrogels show tunable mechanics with self-healing and stress relaxation**

To evaluate the mechanical properties of Trpzip hydrogels, we performed in situ oscillatory parallel plate rheometry on gels at various peptide concentrations. The stiffness of Trpzip hydrogels increased with peptide content, with $G'$ values ranging from 1–60 kPa and all gels reaching an equilibrium storage modulus after approximately 12 h (Fig. 3a). Trpzip gels display temperature-dependent gelation behavior, with a ten-fold higher stiffness at 37 °C compared to 20 °C (Fig. 3b). Surprisingly, the yield point of Trpzip gels occurs at a shear stress of approximately 75 Pa (Fig. 3c). This is consistent with our observations of rapid fluidization of the gel under moderate force (Supplementary Video 1). The Trpzip gel also shows shear-thinning behavior (Fig. 3d), with viscosity decreasing linearly in proportion to the shear rate, suggesting it may be an ideal biomaterial for extrusion and biofabrication. Since other peptide hydrogels can reversibly assemble and disassemble their physically crosslinked networks[22], we performed a thixotropic test on Trpzip gels to determine how fast they self-heal and recover after shear. After exposure to 5% strain for 5 min, the hydrogels re-crosslink immediately upon cessation of shear forces, returning to the initial stiffness within an hour (Fig. 3e). A frequency

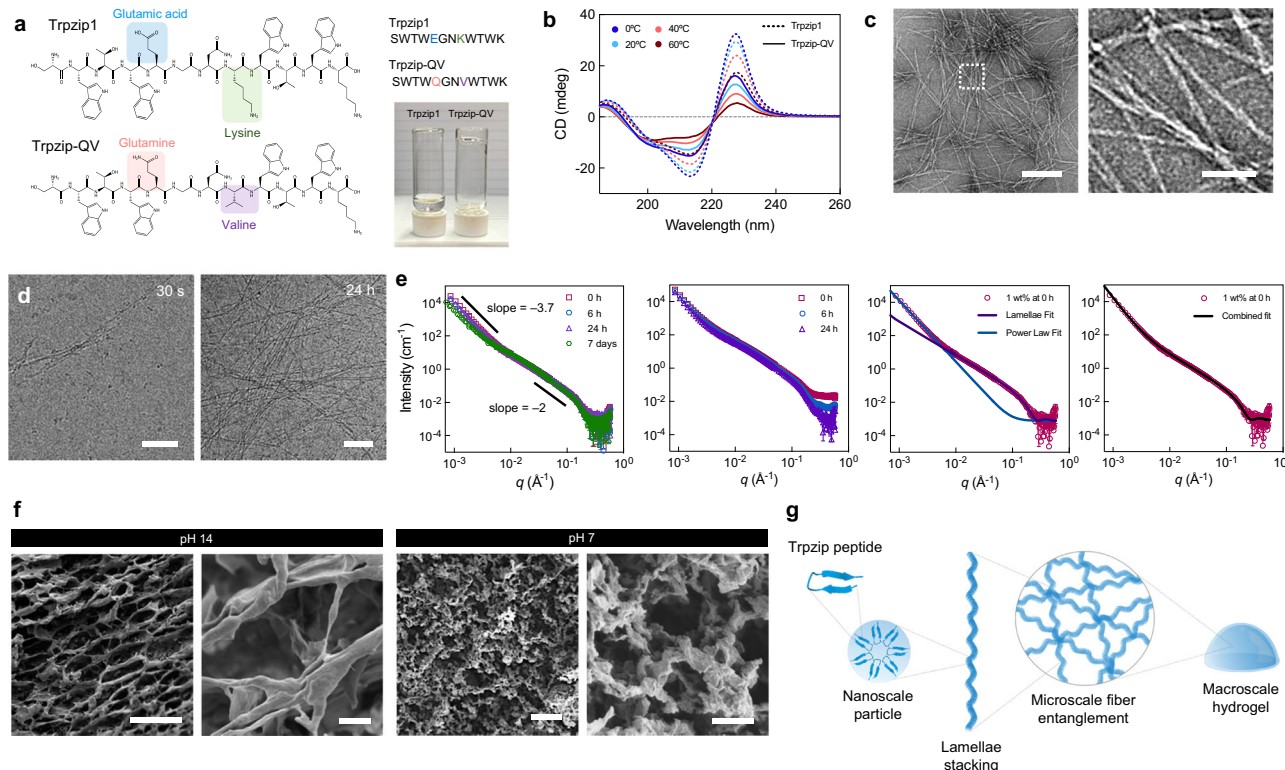

**Fig. 2 | Synthesis, optimization, and characterization of Trpzip hydrogels.**
**a** Chemical structures of Trpzip1 and Trpzip1-QV, with changes in amino acid sequence highlighted. The photograph depicts Trpzip1 and Trpzip-QV peptides dissolved in pure water at 1 mg/mL after 24 h at 37 °C. **b** Circular dichroism spectra of Trpzip1 (dashed line) and Trpzip-QV (solid line) in H₂O across a temperature range of 0–60 °C. **c** Transmission electron micrographs of Trpzip-QV nanofibers formed at pH 7. The scale is 200 nm (left) and 50 nm (right). Data is representative of two independent experiments. **d** Cryo-TEM of Trpzip-QV hydrogels (2% w/v in DMEM, pH 7) at 30 s and 24 h post-gelation at 37 °C. The scale is 50 nm (left) and 100 nm (right). Data is representative of two independent experiments. **e** Small angle neutron scattering profiles of Trpzip-QV hydrogels in deuterated DMEM at 1% w/v (far left), 3% w/v (center left), separate data fitting to the lamellae model fit, and the power law fit (center right), and the combined data fitting against the experimental scattering of 1% w/v Trpzip-QV hydrogels (far right). **f** Scanning electron micrographs of Trpzip-QV hydrogels (3% w/v, DMEM) at pH 14 and pH 7. Scale bars are 100 μm, 5 μm, 200 nm, and 10 μm, from left to right. Data is representative of two independent experiments. **g** Schematic of proposed self-assembly mechanism of Trpzip-QV peptide monomers into hydrogels.

amplitude sweep test confirmed the frequency used for all oscillatory tests (1 Hz) was well within the linear viscoelastic region (Supplementary Fig. 9). Additionally, we found Trpzip gels could also be prepared in pure water, with gelation triggered through the addition of physiological buffers or cell culture media upon mixing. Interestingly, the stiffness of gels formed with this method at half the concentration (0.5% w/v) approximated the stiffness of gels formed using the pH trigger method (1% w/v; Supplementary Fig. 10).

One of the important aspects of natural tissues is their ability to show non-linear viscoelastic properties like stress-relaxation, essential in distributing cellular forces generated during tissue expansion and morphogenesis. To assess whether Trpzip gels possess a similar stress relaxation profile, we performed constant shear deformation measurements (Fig. 3f) and compared the stress relaxation half-time of Trpzip gels (1% w/v) to the natural matrix Matrigel and polyethylene glycol dimethacrylate (PEG-DM) hydrogels. Unsurprisingly, PEG-DM exhibited little stress relaxation due to its covalently cross-linked network; however, due to their dynamic nature, Trpzip gels show a similar stress-relaxation profile to Matrigel. We found the stress relaxation half-time of Trpzip gels to be lower (58.5 ± 0.4 s) compared to Matrigel (87.1 ± 0.76 s) (Fig. 3g), but within the range of other natural matrices[23]. Another important aspect of natural materials is the presence of cell adhesion motifs. We examined the effect of appending the laminin-derived Ile-Lys-Val-Ala-Val (IKVAV) adhesion motif onto our Trpzip hydrogels. Integrating Trpzip–IKVAV (200 μM) into a Trpzip gel resulted in no statistically significant change in stiffness compared to pure Trpzip gels (Fig. 3h). This

suggests that decoupling materials parameters such as stiffness and adhesivity in Trpzip gels is possible.

## Trpzip hydrogels are antimicrobial and support mammalian cell growth

We next investigated the ability of Trpzip hydrogels to support mammalian cell growth. The low yield point of Trpzip gels enables cells to be easily resuspended after adjustment to neutral pH, circumventing the need for large pH switches common to peptide-based hydrogels[24]. We encapsulated human fetal fibroblasts (HFFs) in Trpzip hydrogels and observed their viability after five days to be comparable to cells grown on tissue culture plastic (Fig. 4a, b). Cells exhibited spreading and elongation in Trpzip hydrogels in the absence of exogenous ECM proteins or adhesive peptide motifs, likely due to the positively charged nature of the material. To explore this further, we encapsulated HFFs in both Trpzip gels and Fmoc-GFF gels, another bioinert peptide hydrogel (Fig. 4c)[25]. Cells grown in Trpzip and Fmoc-GFF gels were initially similar in size on day 1, however, cells in Trpzip gels doubled in area after 7 days compared to those grown in Fmoc-GFF gels (Fig. 4d; 23.7% increase). Cellular aspect ratio and cellular roundness also increased a greater amount for cells grown in Trpzip gels compared to Fmoc-GFF gels by day 7 (Supplementary Fig. 11a, b). Strikingly, cells in Trpzip gels show similar morphology to cells cultured in Matrigel after seven days (Supplementary Fig. 12a–d), and the hydrogel itself remains completely intact after one month of culture (Fig. 4e; Supplementary Fig. S13a–c). In addition, we cultured myoblast progenitor cells (C2C12s) in Trpzip gels for one month, and similarly

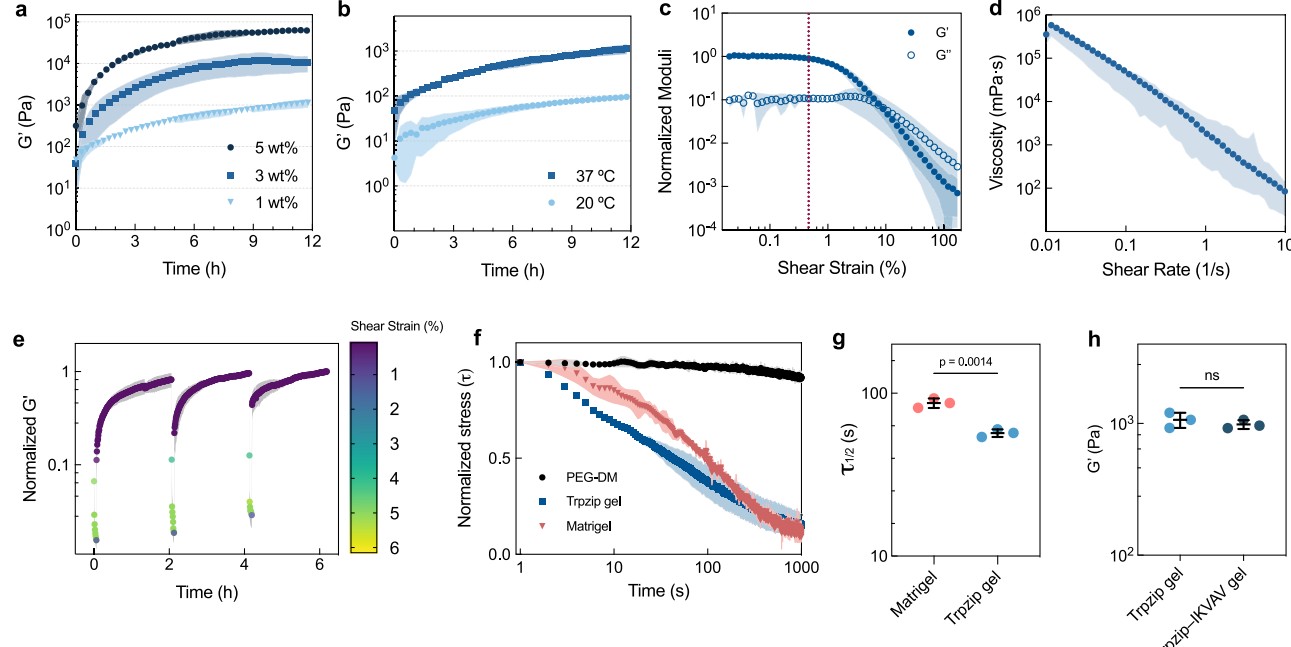

**Fig. 3 | Mechanical characterization of Trpzip hydrogels. a** Oscillatory time sweeps of Trpzip-QV hydrogels of varied concentration represented in percent weight per volume (% w/v). Data is shown as mean ± s.d. (shaded area) from $n = 3$ independently prepared gels. **b** Oscillatory time sweeps of Trpzip-QV hydrogels (1% w/v, DMEM, pH 7) at 20 °C and 37 °C (light and dark blue, respectively). Data is shown as mean ± s.d. (shaded area) from $n = 3$ independently prepared gels. **c** A strain sweep was performed at 1 Hz of a Trpzip-QV hydrogel (1% w/v, DMEM, pH 7). The dotted pink line indicates yield-point. Data is shown as mean ± s.d. (shaded area) from $n = 3$ independently prepared gels. **d** A viscosity flow curve of Trpzip-QV hydrogels (1% w/v, DMEM, pH 7). Data is shown as mean ± s.d. (shaded area) from $n = 3$ independently prepared gels. **e** Thixotropic test on Trpzip-QV hydrogels (3% w/v, DMEM, pH 7) involving exposure to 5% shear strain for 5 min,

followed by reduction of shear strain to 1% for 2 h, repeated three times in succession. Data is shown as mean ± s.d. (shaded area) from $n = 3$ independently prepared gels. **f** Stress-relaxation profiles of Trpzip-QV hydrogels (1% w/v, DMEM, pH 7), Matrigel, and polyethylene-glycol dimethacrylate (PEG-DM) hydrogels. Data is shown as mean ± s.d. (shaded area) from $n = 3$ independently prepared gels. **g** Relaxation half-time ($\tau_{1/2}$) of Trpzip-QV hydrogels (1% w/v, DMEM, pH 7) and Matrigel. *P*-values were calculated using a two-tailed unpaired *t*-test. Data is shown as mean ± s.d. from $n = 3$ independently prepared gels. **h** Effect of IKVAV-Trpzip-QV peptide (200 μM concentration) on bulk hydrogel stiffness. Data is shown as mean ± s.d. from $n = 3$ independently prepared gels. *P*-values were calculated using a two-tailed unpaired *t*-test.

observed evidence of continued proliferation throughout the original gel volume but no dissolution of the material (Supplementary Fig. 13b, d). Through immunofluorescence imaging, we observed the deposition of collagen I throughout Trpzip gels after 14 days of fibroblast culture, suggesting endogenous deposition of ECM is occurring (Fig. 4f; Supplementary Fig. 14a, b). Taken together, these data indicate Trpzip hydrogels foster robust cell attachment, spreading, and elongation without the need for cell adhesion cues. We note most other synthetic peptide-based hydrogels lack this inherent bio-adhesivity[26], suggesting Trpzip gels may prove a versatile 3D cell culture material.

The low yield point and self-healing properties of Trpzip gels also open the potential for syringe delivery. In liquid cell suspensions, mechanical damage results in a significant loss of viability[27]. We reasoned that Trpzip gels may be able to shield cells from the damaging mechanical forces experienced during flow. To assess this, HFFs encapsulated in Trpzip gels were extruded through a syringe needle at high shear. After 24 h, the sheared cell viability was comparable to cells grown on glass and non-extruded cells encapsulated in Trpzip gels (Fig. 4g). The yield-stress fluid properties of Trpzip gels also provide scope for use as a support medium for deposition of material; that is, the ability to fluidize under shear and self-heal around a deposited material. To demonstrate this, we printed high-density fibroblast cell inks into various droplet and line constructs within a support bath of Trpzip gel (Fig. 4h; Supplementary Fig. 15). Dot diameters averaged 650 ± 80 μm ($n = 8$), and line diameters averaging 350 ± 50 μm ($n = 4$; 14% from theoretical line width). Immunofluorescence imaging of the printed constructs shows they consist of tightly packed cells (Fig. 4h), suggesting that

the yield point is high enough for use as a support medium for the printed deposition of live cells.

The amino acid tryptophan has been shown to possess antibacterial activity through its ability to permeabilise bacterial membranes, thereby causing bacterial cell death[28–30]. We hypothesized that the high tryptophan content in Trpzip peptides may confer some antibacterial activity. To test this, we challenged sterilised Trpzip hydrogels with both a Gram-positive and Gram-negative strain of bacteria (*Staphylococcus aureus* and *Escherichia coli*, respectively) and assessed antimicrobial activity after 24 h using a bacterial growth inhibition assay (Fig. 4i; Supplementary Fig. 16). We observed a marked reduction in bacterial growth for both strains, with Trpzip hydrogels showing particularly high activity against *S. aureus* (99.99% reduction in growth for *S. aureus*, 99.75% reduction in growth for *E. coli*). The dual activity of Trpzip gels against both Gram-positive and Gram-negative strains suggests they are promising therapeutic candidates against polymicrobial infections[31]. Like other injectable antimicrobial hydrogels[32,33], Trpzip-based materials may prove useful as extrudable self-healing antimicrobial hydrogels, with potential for use in biomedical applications.

## Trpzip hydrogels can induce polarity changes in human intestinal organoids

Organoid culture relies heavily on Matrigel as an exogenous extracellular matrix support material, despite being of murine origin and having issues with batch variability. We reasoned that Trpzip gels, having similar porosity and stress-relaxing characteristics to Matrigel, could serve as a synthetic, minimally supportive matrix alternative for

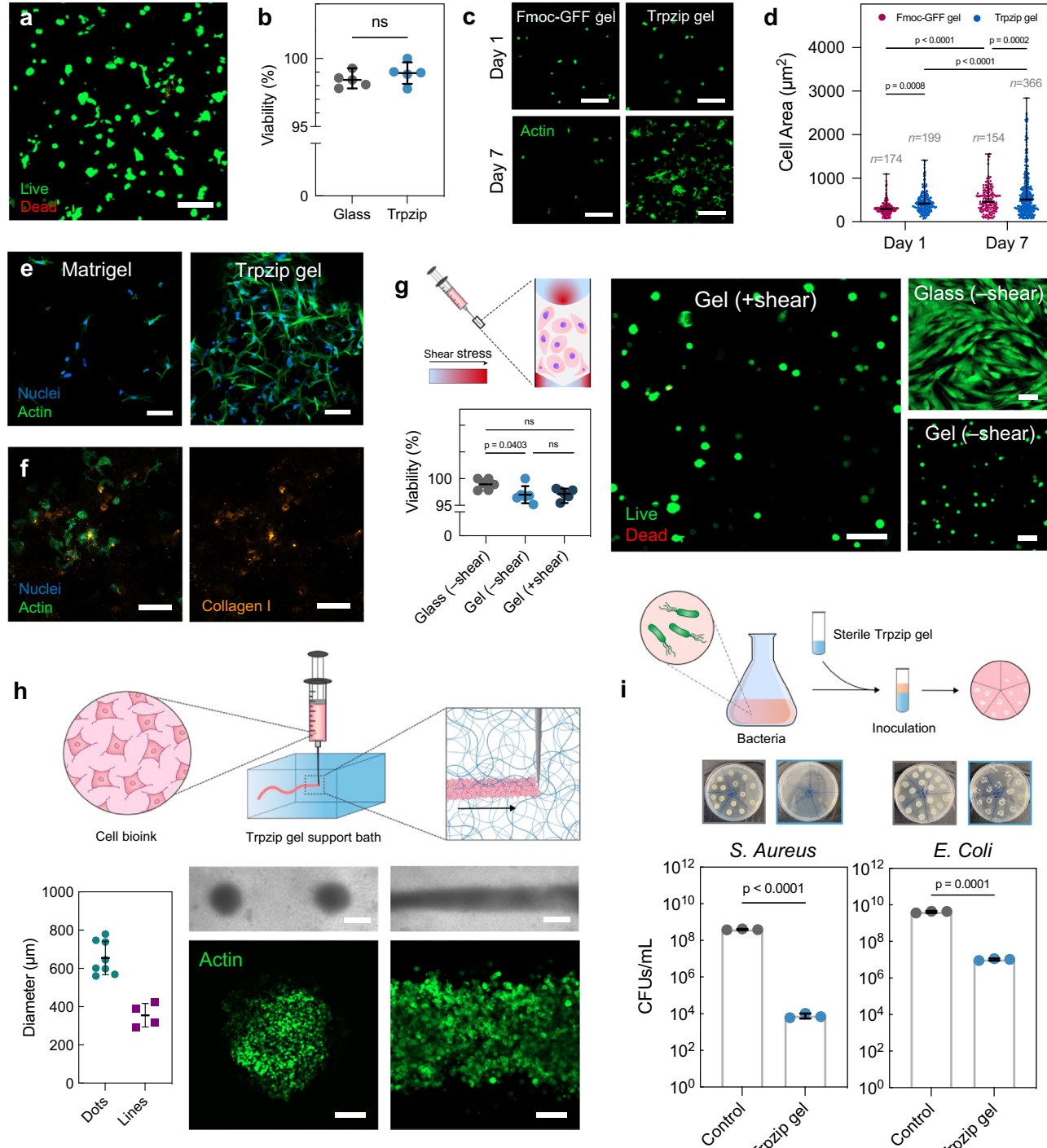

**Fig. 4 | Trpzip hydrogels support cell growth, syringe extrusion, biofabrication and show antimicrobial properties. a** Confocal microscopy image of human fibroblast cells stained with Calcein AM and ethidium homodimer in Trpzip hydrogels (1% w/v, DMEM, pH 7) after 5 days in culture. The scale is 100 µm. **b** Quantified percentage of cell viability of fibroblasts cultured in Trpzip hydrogels compared to culture on glass ($n = 5$ gels). Data are presented as mean ± s.d. *P*-values calculated using two-tailed unpaired *t*-test. **c** Confocal microscopy image of fibroblasts cultured in Fmoc-GFF hydrogels (1% w/v) and Trpzip hydrogels (1% w/v) after seven days, without the use of RGD binding ligands. The scale bar is 50 µm. **d** Quantification of cell area (left) and cell aspect ratio (right) for fibroblasts cultured in Fmoc-GFF hydrogels ($n = 3$) and Trpzip hydrogels ($n = 3$) for seven days. Error bars show mean ± s.d. *P*-values calculated using two-way ANOVA. **e** Confocal microscopy image of HFFs cultured in Matrigel and Trpzip gels (no RGD) for 30 days. The scale bar is 100 µm. Data is representative of one independent experiment. **f** Deposition of endogenous collagen I by fibroblast cells after 14 days

in culture in Trpzip hydrogels without RGD ligands. The scale bar is 100 µm. Data is representative of one independent experiment. **g** Viability of cells encapsulated in Trpzip hydrogels and then exposed to shear ($n = 3$ gels) compared to cells experiencing no shear seeded on glass (2D; $n = 3$ wells) or within Trpzip gels (3D; $n = 3$). Error bars show mean ± s.d. *P*-values calculated using one-way ANOVA. The scale is 100 µm. **h** Bioprinted constructs of cells in a Trpzip hydrogel support bath, either as droplets (left panel) or lines (right panel). The scale is 500 µm in optical images and 200 µm in immunofluorescence images. The measured average diameter of printed cellular droplets ($n = 8$) or lines ($n = 4$). Error bars show mean ± s.d. **i** Antibacterial activity of Trpzip hydrogels against both a gram positive (*S. aureus*) and a gram negative (*E. coli*) species, assessed using a bacterial growth inhibition assay. Photographs show agar plates used to count colony forming units (CFUs) per mL of each bacterial species after incubation with Trpzip gels for 24 h at 37 °C ($n = 3$ gels). Data are presented as mean ± s.d. *P*-values calculated using two-tailed unpaired *t*-test.

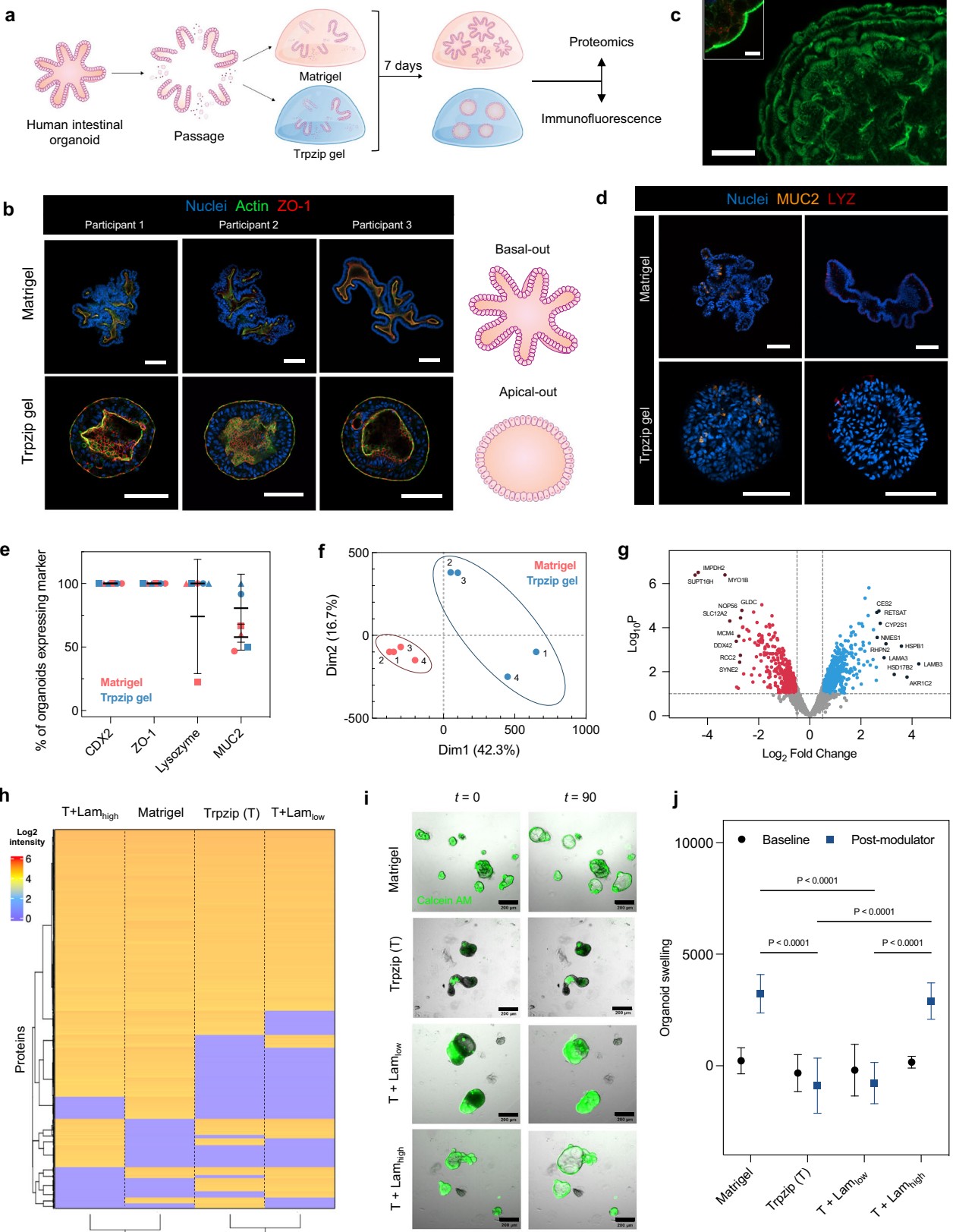

organoid growth. We selected human intestinal organoids—derived from adult stem cells and induced pluripotent stem cells (iPSCs)—as their morphogenesis and tissue patterning pathway have been extensively studied[34–36].

We passaged and seeded intestinal organoids from three participants with cystic fibrosis in Trpzip gels and cultured them for seven days before comparing them alongside organoids grown in Matrigel (Fig. 5a). The viability of organoids was confirmed after seven days using Calcein AM and ethidium homodimer (Supplementary Fig. 17a, b). Organoids grown in Matrigel had projections and budding, with a centralized lumen, whereas organoids grown in Trpzip hydrogels displayed a more spherical morphology (Fig. 5b). Morphometric analysis

**Fig. 5 | Development of adult-stem cell derived intestinal organoids in Trpzip hydrogels compared to Matrigel. a** Schematic describing experimental design to evaluate organoid development in Trpzip hydrogels compared to Matrigel. **b** Representative images across three independent experiments of organoid morphology after 7 days in Matrigel and Trpzip gels across three participant organoid lines. Scale is 100 μm (top panel); 50 μm (bottom panel). **c** Filamentous actin stain to highlight microvilli localization on the exterior of organoids grown in Trpzip gels (scale is 100 μm). The inset shows microvilli from a cross-section view of the organoid (scale is 50 μm). Data is representative of three independent experiments. **d** Representative images across three independent experiments of organoids grown in either Trpzip gels or Matrigel marked for differentiated cell types. Paneth cells and goblet cells are marked by LYZ and MUC2 respectively. The scale is 100 μm. **e** Frequency of mature cell type differentiation in Trpzip gels (blue) and Matrigel (red). Data points represent the percentage of organoids expressing each marker as assessed by antibody staining across three participant organoid lines

with $n = 10$ organoids per marker. Error bars show mean ± s.d. **f** Principal component analysis of protein expression profiles of organoids cultured in Matrigel and Trpzip gels. **g** Volcano plot indicating significantly differentially expressed proteins. The dashed horizontal line shows $P$-value cut-off and vertical lines indicate up/down-regulated proteins. $P$-values were calculated using the Benjamini-Hochberg procedure. **h** Hierarchical clustering analysis of global proteomics for organoids cultured in Matrigel, pure Trpzip, and Trpzip with low and high laminin content. Clustering distance is based on the Euclidean distance. **i** Brightfield live imaging over 90 min of organoid response to treatment with the CFTR modulator (Trikafta) in Matrigel, pure Trpzip, and Trpzip with low and high laminin content. Organoids have been stained with Calcein AM (green) to visualize viability. The scale is 200 μm. **j** Quantification of change in organoid size (normalized AUC $t = 90$; $n = 5$ organoids per condition) following response to CFTR modulator (blue) compared to baseline (black). Data are presented as mean ± s.d. $P$-values calculated using two-way ANOVA.

revealed an average circularity value of 0.421 in Matrigel-grown organoids and 0.807 in Trpzip-grown organoids, across the three participant lines (Supplementary Fig. 18). In contrast to Matrigel-grown organoids, we observed Trpzip-grown organoids possessed an apical out polarity, as evidenced by a layer of filamentous actin and the apical tight-junction ZO-1 protein on the organoid exterior, along with the visualization of microvilli of brush border cells (Fig. 5b, c; Supplementary Fig. 19a, b)[37]. We observed the same reversal of polarity in iPSC-derived small intestinal organoids, as evidenced by filamentous actin imaging (Supplementary Fig. 20).

To understand the differentiated state of these intestinal organoids, we performed further immunofluorescence analysis. Organoids grown in both Matrigel and Trpzip gels were examined for the presence of intestinal epithelial patterning and polarization, as well as markers of fully differentiated intestinal cell types including Paneth cells and goblet cells (Fig. 5d). Immunofluorescence analysis revealed the expression of the intestinal epithelial marker CDX2 and the tight junction marker ZO-1 in 100% of organoids, irrespective of participant origin or matrix condition, confirming Trpzip gels support intestinal lineage differentiation and proper epithelial polarization (Fig. 5e). We also quantified the percentage of organoids expressing the markers MUC2 and LYZ to assess abundance of the differentiated goblet cells and Paneth cells, respectively. We found Matrigel-grown and Trpzip-grown organoids express all markers with a similar frequency, although variability between participant lines was evident. We then performed global proteomics for organoids cultured in Matrigel and Trpzip hydrogels, which demonstrated 70% similarity with separate clustering behavior by principal component analysis for the different participants (Fig. 5f). Differential expression analysis indicates changes in proteins involved in intestinal homeostasis and metabolism (Fig. 5g), with increased cytoskeletal remodeling evident in Trpzip grown organoids (Supplementary Fig. 21). These expression differences are unsurprising considering the different morphology and apparent apical polarity fostered by the hydrogels.

It is clear the surrounding matrix in organoid culture can play an important role in regulating growth, polarity, and morphogenesis. Most protocols use Matrigel as the encapsulate; however, the use of alternative biomaterials has been of great interest in recent years. Early iterations of synthetic matrices were often PEG-based, with the incorporation of MMP-degradable peptides[36,38] to enable organoid morphogenesis contingent on matrix degradation but have also included polysaccharide and other full-length protein-based hydrogels[39,40]. Recently, Lutolf and colleagues showed how a fully synthetic hydrogel with both covalent and physical crosslinks enabled comparable organogenesis to Matrigel by virtue of matrix stress-relaxation[23]. Trpzip gels show strong potential as a fully synthetic organoid matrix, emulating both the stress-relaxing nature and dynamic hierarchical assembly observed in natural materials.

Materials that foster apical-out polarity would benefit a range of studies since this is the side that natively interacts with the external environment. Most techniques for accessing the apical surface involve invasively penetrating the organoid barrier through either micro-injection, shearing, or mechanical disruption of organoids, or dissociation of 3D organoids to be reseeded as 2D monolayers on Transwell plates. Co et al. only recently reported the first non-destructive 3D method of generating apical-out organoids through suspension culture[37]. However, this approach precludes studies of the epithelial microenvironment, an aspect that can be tuned using Trpzip hydrogels. Once the Trpzip-embedded organoids grow larger (>100 μm), there is evidence for both outward and inward polarization with an internal lumen. This characteristic will not impede studies of apical facing organoid behavior.

Previous work has demonstrated how the presence of laminin protein is important for supporting basal polarity and growth in synthetic hydrogels[41]. While apical facing organoids hold several advantages as discussed, we next sought to evaluate whether the addition of laminin would influence the polarity and growth characteristics of Trpzip-encapsulated organoids. We blended Trpzip hydrogels with varying amounts of laminin to determine whether we could mimic the polarity and growth characteristics of organoids grown in Matrigel. We cultured organoids from one participant in Matrigel, pure Trpzip gels, Trpzip gels incorporating 0.4 mg/mL laminin (low), and Trpzip gels incorporating 3 mg/mL laminin (high). Organoids grown in pure Trpzip gels and those containing low laminin maintained a spherical morphology (Supplementary Fig. 17a). In contrast, organoids grown in the higher laminin content Trpzip gels appeared more similar in morphology to organoids grown in Matrigel. Live/dead imaging indicates a slight increase in cell death for the Trpzip-only conditions, which we attribute to normal apoptotic signaling during morphogenesis as observed previously with apical-out organoid protocols (Supplementary Fig. 17b)[37]. The addition of laminin also increases organoid viability in the Trpzip hydrogels. Global proteomics analysis shows significant similarity between all conditions – pure Trpzip, Trpzip with low and high laminin, and Matrigel (Fig. 5h). Importantly, we observe clustering of Trpzip with Trpzip-low laminin and Matrigel with Trpzip-high laminin, suggesting greater similarity to Matrigel as we increase the laminin content in the matrix (Fig. 5h; Supplementary Fig. 22a, b).

Since the addition of laminin to Trpzip appears to emulate growth in Matrigel, we performed a forskolin swelling assay to quantitatively assess organoid polarity. As the organoids were derived from patients with the homozygous DF508-CFTR mutation, organoids with a basal-out polarity are expected to swell upon correction and activation of the cystic fibrosis transmembrane conductance regulator (CFTR) channel. Upon activation of the CFTR channel, organoid swelling was observed within 90 min with live imaging (Fig. 5j, k; Supplementary Video 2–5) in organoids grown in Matrigel and Trpzip-high laminin hydrogels. Conversely, organoids grown in Trpzip gels and Trpzip-low

laminin exhibited no such swelling, suggesting the organoids grown in these matrices display apical-out polarity. This is confirmed by the lack of swelling that is also observed when organoids are grown in suspension, a process by which is known to cause a reversion to apical-out polarity (Supplementary Fig. 23)[37]. Taken together, we show basal-out polarity can be maintained in organoids grown in Trpzip hydrogels when a sufficient concentration of laminin protein is present.

Our work demonstrates a peptide hydrogelator based on the tryptophan zipper motif that self-assembles into a nano- and microstructured material with unique mechanical and biological properties. Trpzip hydrogels are easily formed without the need for rigid temperature control, which is often required for natural matrices. The tunable modulus and low yield stress provide the first example of a material where viscoelasticity can be varied to direct functional biological outcomes followed by quick harvest through simple agitation. This will prove highly beneficial for molecular characterization, which usually requires invasive enzyme-mediated dissolution of the surrounding matrix. Similarly, the low yield stress and self-healing properties provide a means for syringe extrusion, where fluidization of the hierarchical material protects the cells from shear, towards applications in cell delivery and in biofabrication. Considering how these hydrogels are simultaneously bactericidal and bioactive to mammalian cells, there is broad scope for using Trpzip hydrogels in vitro as well as in vivo as a therapeutic biomaterial.

## Methods

### Materials
Synthetic peptides (Trpzip-QV and Trpzip-IKVAV) were purchased as the formate salt (purity >98%) from GenScript Biotech (Singapore) Pte Ltd and used without further purification.

### Molecular dynamics
Avogadro 1.2.0 was used to create all-atom peptide structure files. The MARTINI force field (version 2.2) and martinize.py[42] were used to generate the coarse grain (CG) form of each peptide variant. A cubic simulation box ($20 \times 20 \times 20$ nm$^3$) was created using GROMACS version 2021.3[43], and filled with 150 zwitterionic CG peptide monomers (final peptide concentration of 31 mmol L$^{-1}$). The system was maintained at neutral pH by using amino acid side chains set to their standard charge states at pH 7. Periodic boundary conditions were applied in all three dimensions. Non-polarizable CG water was used to solvate the simulation box, and counterions Na$^+$ and Cl$^-$ were added if needed to neutralize the overall charge of the system. Electrostatic interactions and Lennard-Jones interactions were shifted to zero at a cutoff of 1.1 nm. Electrostatic interactions were screened by setting the relative dielectric constant ($\varepsilon_r$) to 15. The system was energy minimized using the steepest descents algorithm for 10,000 steps or until forces in excess of 100 kJ mol$^{-1}$ nm$^{-1}$ were removed. The Berendsen thermostat[44] was used to maintain temperature ($\tau_T = 1$ ps) and pressure ($\tau_P = 3$ ps) around 298 K and 1 bar, respectively. The LINCS constraint algorithm[45] was used to fix bond lengths.

Scripts used to run CG simulations were based on the Martini tutorials, 'High throughput peptide self-assembly', available online at http://cgmartini.nl/index.php/tutorials-general-introduction-gmx5/tutorial-ht-peptide-gmx5. Coarse-grain structures were converted to all-atom resolution using backward.py[46].

### Peptide synthesis and purification
Peptides were synthesized using solid phase peptide synthesis on a Biotage Initiator+ Alstra microwave peptide synthesizer. Briefly, 2-chlorotrityl chloride resin (1.1 mmol/g loading; 500 mg; Chem-Impex) was allowed to swell in dichloromethane (DCM) for 30 min. The first amino acid (3 equivalents) was dissolved in a solution of 2 mL DCM, 2 mL peptide-grade N,N-dimethylformamide (DMF) and 1 mL

N,N-diisopropylethylamine (DIPEA), before being allowed to couple to the resin overnight. Methanol capping was performed once after the first amino acid coupling. Fluorenylmethoxycarbonyl (Fmoc) deprotection was performed using 20% piperidine in DMF (repeated twice, once for 5 min and again for 10 min). Subsequent amino acids were coupled using benzotriazol-1-yloxytripyrrolidinophosphonium hexafluorophosphate (PyBOP) activation, whereby 3 equivalents of the Fmoc-amino acid, 3 equivalents of PyBOP in peptide-grade DMF, and 6 equivalents of DIPEA were used. Coupling reactions were performed under microwave at 70 °C for 5 min. Following completion of the synthesis, the resin was dried and treated with a cleavage cocktail consisting of 95% trifluoroacetic acid, 2.5% triisopropylsilane, and 2.5% ethanedithiol for 3 h at RT on an orbital shaker. The crude peptide was precipitated dropwise into ice-cold diethyl ether, followed by centrifugation of the mixture (3000 g for 10 min) and decanting of excess diethyl ether to yield a solid pellet of the crude peptide.

For purification, the crude peptide was dissolved in Milli-Q water and HPLC-grade acetonitrile (1:1 ratio) and injected onto an Inertsil ODS-4 C18 column (20 mm I.D., 150 mm length, 5 µm particle size) at a flow rate of 5 mL/min. The eluting solvent system (in all cases with 0.1% v/v formic acid as ion-pairing agent) had a linear gradient of 20% (v/v) acetonitrile in water for 5 min, gradually rising to 80% (v/v) acetonitrile in water at 35 min. This concentration was kept constant until 40 min when the gradient was decreased to 20% (v/v) acetonitrile in water at 42 min. Elution was determined by UV detection at 254 nm. Peptide identity and purity were verified by MALDI-TOF (Supplementary Fig. 2). Purified peptide was collected from HPLC fractions, pooled, and lyophilised to yield a fluffy, white solid.

### Hydrogel preparation
Trpzip hydrogels were prepared by dissolving lyophilized peptide in 1 N NaOH (10% of the final hydrogel volume). The desired cell culture media was added to aid further dissolution of the peptide (50% of the final hydrogel volume). The hydrogel was adjusted to pH 7 using 1 M HCl before additional cell culture media was added to bring the hydrogel to its final desired volume. Hydrogels were allowed to incubate at 37 °C overnight before use in cell culture.

Trpzip hydrogels could also be prepared by dissolving lyophilized peptide in Milli-Q water at a concentration of 1% (w/v). Magnetic stirring on a hot plate (40 °C) was used to accelerate dissolution. Hydrogels were sterilized by exposure to UV light for 15 min. Gelation was triggered by adding cell culture media to a final concentration of 0.5% (w/v).

### Transmission electron microscopy (TEM)
Carbon-coated copper grids (square mesh 300; ProSciTech) were plasma-treated in the air for 1 min. Peptide hydrogels (1% w/v; pH 7) were diluted ten-fold and 5 µL of the resulting solution was pipetted onto the grid. The droplet was left on the grid for 1 min before the excess solution was blotted off with filter paper. The grid was left to air-dry prior to staining. Negative stain (1% aqueous uranyl acetate) was applied for 1 min before excess liquid was blotted off using filter paper. Bright-field TEM micrographs were obtained on an FEI Tecnai G2 TEM operating at an accelerating voltage of 200 kV. Images were acquired using a BM Eagle 2 K CCD camera.

### Circular dichroism (CD)
Peptide solutions were prepared at 0.1 mg/mL in Milli-Q water. Far-UV CD spectra were recorded between 180 and 300 nm on Chirascan Plus CD spectrometer (Applied Photophysics, UK). Spectra was recorded in 1 nm steps and sampled at 0.5 s per point. All measurements were taken in a 1 mm quartz cuvette (Starna, Inc.). Final spectra were recorded as the average of three scans, with baseline spectra subtracted.

## Fourier transform infrared spectroscopy (FTIR)

A Perkin Elmer Spotlight 400 FTIR spectrometer fitted with a diamond crystal attenuated total reflectance (ATR) accessory was used to acquire FTIR spectra. Trpzip hydrogels (3% w/v) were mounted between the diamond crystal and force lever and all spectra were scanned 20 times over the range of 2000–1000 cm$^{-1}$.

## Small and ultra-small angle neutron scattering (SANS and USANS)

Trpzip gels were prepared by dissolving lyophilized peptide in deuterated DMEM before being transported to Australian Centre for Neutron Scattering (ACNS) at the Australian Nuclear Science and Technology Organization (ANSTO). SANS measurements were performed on hydrogels samples at 37 °C.

The structure of the hydrogels was analyzed using the Quokka pinhole small-angle neutron scattering (SANS) instrument[47]. Samples were loaded in the quokka demountable cells of 20 mm diameter and pathlength 1 mm. The scattering data of samples were collected at a full $q$ range of 0.0007-0.73 Å$^{-1}$ with three detector configurations (20 m collimation with 8.1 Å neutrons), 12 m (12 m collimation with 5 Å neutrons) and 1.3 m (12 m collimation with 5 Å neutrons). The scattering profiles of the gels were recorded against the scattering vector by using the following equation:

$$q = \frac{4\pi}{\lambda}\sin\theta \qquad (1)$$

where $q$ is the scattering vector, $2\theta$ is the angle of scattering, $\lambda$ is the wavelength of the neutron beam. The obtained 2D raw scattering data were reduced and converted to the absolute scale using the Igor Pro software package (with Quokka SANS reduction macros)[48]. Igor Pro was used to subtract appropriate solvent background from each dataset. SasView (www.sasview.org) was used to fit the models to the scattering data.

The USANS data were collected using the Kookaburra Bonse–Hart instrument at ACNS, ANSTO[49]. Round demountable cells (40 mm diameter, 1 mm pathlength, 29 mm sample aperture) were used to mount samples. High flux mode was used with a neutron wavelength of 4.74 Å to provide $q$ ranges of $3.5 \times 10^{-5}$–0.007 Å$^{-1}$. The experimental USANS data were de-smeared using the Lake algorithm incorporated in NIST USANS macros. The neutron scattering length densities (SLD) of Trpzip peptides of $1.46 \times 10^{-6}$ Å$^{-2}$ was calculated using the online SLD calculator (https://www.ncnr.nist.gov/resources/activation/). SasView was used to fit the scattering data.

## Scanning electron microscopy (SEM)

Lyophilized hydrogel samples were fixed to aluminum stubs with adhesive carbon discs. The samples and stubs were then sputter coated once with gold (Emitech K575x sputter coater). A Hitachi S3400 (7–10 kV, probe current of 40) scanning electron microscope was used for all imaging.

## Cryo-TEM

Trpzip hydrogels were prepared at 2% (w/v) in DMEM and allowed to gel at 37 °C until the designated imaging time point. Gel (4.5 µL) was applied to glow discharged R2/2 copper grids (Quantifoil Micro Tools) and blotted for 4 s in a controlled humidity chamber (95% humidity), then plunged in liquid ethane using a Lecia EM GP device (Leica Microsystem). The grids were imaged using a Talos Arctica cryoTEM (Thermo Fisher Scientific) and operated at 200 kV, with the specimen maintained at liquid nitrogen temperatures. Images were recorded on a Falcon 3EC direct detector camera operated in linear mode.

## Rheology

Rheological measurements were performed on an Anton Paar MCR 302e Rheometer with a parallel plate geometry (25 mm disc, 1 mm gap height, 560 µL of hydrogel). Oscillatory measurements were performed with 0.2% strain and a 1 Hz frequency at 37 °C, unless specified otherwise. Viscosity flow curve tests were performed with a log ramp up rate from 0.01 to 10 shear rate (1 s$^{-1}$) over 10 min. Strain sweep tests were performed with a log ramp up rate from 0.02% shear strain up to 200% at 1 Hz frequency over 10 min. Stress relaxation tests were performed at 1% strain, after which the stress profile was normalized to the initial maximum stress. The decay half time was calculated by determining the time at which stress was half the initial normalized stress. Frequency sweeps were run with a log ramp up rate from 0.01 to 100 Hz with 0.2% strain (Supplementary Fig. 9).

## Cell and organoid culture

Human fetal fibroblasts (HFF-1; SCRC-1041) were obtained from the American Type Culture Collection (ATCC) and cultured in high-glucose DMEM supplemented with 15% fetal bovine serum (FBS) and 1% penicillin/streptomycin (P/S). All HFF cultures were maintained at 37 °C, 5% $CO_2$ and used between passages 17–22. Mouse myoblast cells (C2C12) were obtained from ATCC (CRL-1772) were cultured in high-glucose DMEM supplemented with 10% FBS and 1% P/S. All C2C12 cultures were maintained at 37 °C, 5% $CO_2$, and used between passages 9–11. For differentiation of C2C12 cells, media consisting of high glucose DMEM supplemented with 10% horse serum and 1% P/S was used. Media changes occurred every 2 days for both cell lines.

Human small intestinal organoids were derived from induced pluripotent stem cells (iPSCs) obtained from ATCC (ACS-1020, lot number 0176) using the STEMdiff™ intestinal organoid kit (StemCell Technologies; cat #05140). All iPSC-derived organoids were maintained at 37 °C, 5% $CO_2$ and in STEMdiff™ intestinal organoid growth medium (StemCell Technologies; cat #05145) and used between passages 12–17. Media changes occurred every 2–3 days, and organoids were passaged every 7–10 days depending on organoid size and density. All cell lines were authenticated by ATCC using STR profiling, as confirmed by the Certificate of Analysis provided upon purchase. iPSCs were additionally karyotyped.

Adult stem cell derived intestinal organoids were sourced from the molecular and integrative Cystic Fibrosis (miCF) Biobank (HREC16/SCH120) which were established from crypts isolated from four to six rectal biopsies[50]. Isolated crypts were seeded in 70% Matrigel (Growth factor reduced, phenol-free; Corning 356231) in 24-well plates at a density of approximately 10–30 crypts in $3 \times 10$ µL Matrigel droplets per well. Intesticult organoid media (StemCell Technologies; cat #06010) was supplemented with 1× penicillin-streptomycin, 50 mg/mL gentamicin, 50 mg/mL vancomycin, 250 ng/mL fungizone and 100 mg/mL primocin. Cryopreserved organoids were seeded in Matrigel and cultured for 7 days with Intesticult organoid media (StemCell Technologies; cat #06010), before being passaged and seeded in equal density in Matrigel or Trpzip hydrogels (with 0.4 mg/mL or 3 mg/mL laminin, designated as low and high laminin content, respectively; R&D Systems Cultrex 3D Culture Matrix Laminin I; RDS344600501). At day 6–7 post-seeding, organoid viability was assessed using Calcein AM (Thermo Fisher Scientific C3100MP), then imaged, fixed, or lysed as necessary.

## Cell viability assay

HFFs were detached from tissue culture flasks with 0.25% trypsin and seeded in Trpzip hydrogels (1 wt%, rehydrated in media) at a concentration of 600,000 cells/mL. The cell laden Trpzip gels were stiffened by incubation at 37 °C for 20 min before cell culture media was added. The media was changed every two days until viability was assessed using a Calcein-AM/Ethidium Homodimer-1 kit (Thermo-Fisher Scientific; cat #L3224). For live/dead staining, Calcein-AM (0.5 µL/mL) and Ethidium-Homodimer-1 (2 µL/mL) were added to

Trpzip gels for 40 min before being washed twice with PBS prior to imaging. For assessing viability of cells after injection, the above procedure was followed except that cell laden Trpzip gels were loaded into a 1 mL syringe before being extruded through an 18 G needle.

## Bioprinting

HFFs were detached from tissue culture flasks with 0.25% trypsin and pelleted via centrifugation (300 g for 3 min). The supernatant was removed and the cell pellet was lightly broken up via pipetting, and care was taken to not introduce air bubbles. The pellet was then drawn into a 1 mL syringe, and the syringe inserted directly into a 3D printed fitting on a Lulzbot 3D bioprinter. The desired syringe needle was then primed with cell solution and cells were printed as either dots or lines into custom 3D-printed square molds (0.8 mm × 0.8 mm) containing Trpzip gel (1% w/v) as a support bath. The printed constructs were then incubated at 37 °C for 20 min to allow stiffening of the gel before fixation with 4% paraformaldehyde (PFA) for immunofluorescence analysis.

## Antimicrobial assays

Briefly, *S. aureus* 38 or *E. coli* K12 were inoculated in 5 mL of Mueller Hinton Broth (MHB) media and incubated at 37 °C to establish an overnight culture. Twenty-four hours post incubation, the bacteria were pelleted out of the MHB via centrifugation at 1000 g for 10 min. An Eppendorf 5430 multifaceted centrifuge was used for centrifugation. The supernatants were subsequently discarded, and the bacteria were dissolved in fresh MHB. The volume of MHB was adjusted such that the solution of bacteria in MHB recorded an absorbance value of 0.1 in a microplate reader (600 nm), corresponding to a starting concentration of the *S. aureus* 38 or *E. coli* K12 in the MHB of $10^8$ bacteria/mL. The media was further diluted to obtain a final bacterial concentration of $10^5$ bacteria/mL before being added to the vials containing sterile Trpzip hydrogel (1% w/v; 1 mL). The vials were then transferred to a humidified incubator and incubated overnight at 37 °C. Post incubation, the media from each vial was serially diluted from 1 to 10-fold in sterile PBS. Subsequently, the diluted media was pipetted out on agar plates. The agar plates were then incubated overnight at 37 °C. The following day, the number of visible single bacterial colonies, referred to as the Colony Forming Units (or CFUs), were counted. The CFUs observed in the agar plate corresponding to the *n*th dilution was then used to calculate the CFUs/mL of the undiluted media using the following formula:

$$\frac{CFUs}{mL}(undiluted\ media) = CFUs\ in\ n^{th}\ dilution \times 10^n \qquad (2)$$

## Organoid encapsulation in Trpzip gels

Organoids were recovered from Matrigel using ice-cold DMEM/F12 basal media (Life Technologies 12634-010) and mechanically fragmented via pipette mixing. Organoids were then pelleted via centrifugation (300 g for 5 min at 4 °C), and the supernatant removed. Organoids were resuspended in Trpzip gel before being added to a plastic 96-well tissue culture plate in 50 μL droplets. The plate was allowed to incubate at 37 °C for 30 min to allow the gel to stiffen before the addition of media.

## Global proteomics

Organoids were extracted after 7 d of culture in either Matrigel or various Trpzip gels, and subsequently lysed using RIPA buffer (50 μL; Life Technologies 89900) and protease inhibitor cocktail (Sigma 11836153001), followed by 15 × 30 s cycles of sonication at 4 °C using the Bioruptor Pico (Diagenode B01060010). The Pierce BCA Protein Assay Kit was used to calculate protein concentrations. Lysates were reduced with 5 mM dithiothreitol (37 °C for 30 min), alkylated using 10 mM iodoacetamide (RT for 30 min) and incubated with trypsin (1:20

w/w; 37 °C for 18 h) before being desalted using two SDB-RPS disks (Empore, Sigma-Aldrich 66886-U). Nano-LC was performed to separate proteolytic peptide samples using an UltiMate nanoRSLC UPLC and autosampler system (Dionex, Amsterdam, Netherlands). Analysis was performed using an Orbitrap Fusion Lumos Tribrid mass spectrometer (Thermo Scientific, Bremen, Germany).

MaxQuant (version 2.1.3), coupled with the Andromeda algorithm, was used to analyze raw peak lists. The search parameters were as follows: ±0.5 Da for peptide fragments; ±4.5 ppm tolerance for precursor ions; oxidation (M) and N-terminal protein acetylation as variable modifications; carbamidomethyl (C) as a mixed modification; and enzyme specificity as trypsin with two missed cleavages possible. The human Swiss-Prot database (August 2022 release) was used to search peaks. The MaxLFQ algorithm (using default parameters) was used for label-free protein quantification. The R package, Differential Enrichment analysis of Proteomics data (DEP), was used for downstream processing of quantified proteomics data. Only proteins present in at least 50% of the samples in each of Matrigel encapsulated and Trpzip gel samples were used for subsequent analysis. Differential expression analysis was carried out using a log fold change cutoff of 0.264 and *p*-value cutoff of 0.05. The QIAGEN IPA Fall Release 2022 (30 September 2022) was used to perform canonical pathway analysis of differentially abundant proteins. The mass spectrometry proteomics data have been deposited to the ProteomeXchange Consortium via the PRIDE partner repository with the dataset identifier PXD038870.

## Fixation and immunofluorescence staining

All organoids were fixed and immunostained following the protocol described here[51]. In more detail, organoids in Matrigel domes were incubated on ice using ice-cold cell recovery solution for 20 min to remove excess Matrigel. Organoids were washed once more in fresh cell recovery solution before being fixed in 4% methanol-free PFA for 45 min at 4 °C. Organoids in Trpzip gels were collected by pipetting the gel up and down to release the organoids, before transferring the organoids and gel to a 15 mL tube on ice. Excess peptide was removed by diluting the mixture 3-fold with cold basal media, then allowing organoids to sink via gravity before aspirating excess peptide aggregates that remained in suspension. This process was repeated until all peptide was removed, as indicated by the increasing clarity of the media used to rinse the organoids. Organoids in Trpzip gels were then fixed in 4% methanol-free PFA for 45 min at 4 °C. Excess PFA was then removed, and organoids were permeabilised with 0.1% (v/v) tween-20 in PBS (10 min at 4 °C). Fixed organoids were transferred to a nonadherent 24-well plate for the remainder of the immunostaining process. Blocking was performed using 0.1% organoid washing buffer (OWB; 1% bovine serum albumin and 0.1% triton X-100 in PBS) for 20 min at RT. Primary antibodies were diluted in OWB to their working concentration and incubated with fixed organoids overnight at 4 °C. Fixed organoids were then washed in OWB at least 3 times over the span of 1 d before secondary antibodies were diluted in OWB to their working concentration and incubated overnight at 4 °C. Fixed organoids were washed again in OWB at least 3 times over the span of 1 d before being cleared with a fructose-glycerol clearing solution (60% v/v glycerol and 2.5 M fructose) for 20 min at RT. Immunostained organoids were then mounted onto a microscope slide in the fructose-glycerol clearing solution for imaging.

## Microscopy

All fixed and immunostained cell and organoid samples were imaged using a Zeiss LSM 800 confocal microscope and Zen Blue software unless specified otherwise.

## Forskolin-induced swelling assay

After 6–7 days of culture in Matrigel or various Trpzip gels, organoids less than 100 μm in size were selected and seeded in 4 μL Matrigel

droplets in a 96-well plate. For viability assessment, organoids were incubated with Calcein AM (5 μM; Thermo Fisher Scientific C3100MP) for 30 min prior to addition of forskolin (5 μM). For cystic fibrosis transmembrane conductance regulator (CFTR) targeted correction, Trikafta (VX-770 + VX-661 + VX-445) was added together with forskolin. Organoid swelling was monitored using time-lapse brightfield imaging (acquired at 10 min intervals for 90 min) using a Nikon Eclipse Ti2-E microscope coupled with a Andor Zyla 4.2 sCMOS camera. A custom-built script was used to quantify organoid swelling. Total organoid surface area post-forskolin exposure was calculated and normalized against initial organoid surface area to determine the relative amount of swelling. The area under the curve (calculated increase in organoid surface area from $t = 0$ to $t = 90$; baseline=100%) was then calculated using GraphPad Prism.

### Reporting summary

Further information on research design is available in the Nature Portfolio Reporting Summary linked to this article.

## Data availability

The datasets generated and/or analyzed in this study are available in the ProteomeXChange Consortium under accession code PXD038870, and from the corresponding author upon request.

## Code availability

The scripts used for MD simulations can be found on Github (https://github.com/Ash-Nguyen/Trpzip).

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

## Acknowledgements

A.K.N. acknowledges scholarship support from the Australian Government Research Training Program and the Baxter Family Postgraduate Scholarship. This work was supported through funding from the Australian Research Council Grant FT180100417 (K.A.K.), the National Health and Medical Research Council Grant APP1185021 (K.A.K.) and APP1188987 (S.A.W.), the National Cancer Institute of the National Institutes of Health Grant R01CA251443 (K.A.K.), and the Sydney Children Hospital Network Foundation (S.A.W.) and Luminesce Alliance Research grants (S.A.W.). We thank the study participants and their families for their contributions. We also thank Sydney Children's Hospitals (SCH) Randwick Cystic Fibrosis clinic especially Prof Adam Jaffe, A/Prof Keith Ooi, Dr Laura Fawcett, Dr Yvonne Belessis, Leanne Plush, Amanda Thompson and Rhonda Bell in the organization and collection of participant biospecimens for miCF biobank. The authors acknowledge the help and support of staff at the Katharina Gaus Light Imaging Facility (KGLMF) of the UNSW Mark Wainwright Analytical Centre. The authors acknowledge the use of the Cryo Electron Microscopy Facility through the Victor Chang Cardiac Research Institute Innovation Centre, funded by the NSW government and the Electron Microscope Unit at UNSW Sydney. This study used the computational cluster Katana supported by Research Technology Services at UNSW Sydney. We would also like to thank the Australian Nuclear Science and Technology Organisation (ANSTO) for providing USANS and SANS beam facilities under proposal number P 14142 for this work.

## Author contributions

A.K.N. and K.A.K. conceived and initiated the study. A.K.N. carried out experiments across the entire range of approaches, analyzed data, prepared figures, and organized the preparation of the manuscript. T.G.M. performed SEM imaging, stress-relaxation rheology measurements, 3D bioprinting, cell morphometric analyses, and advised rheological measurements. E.K. and S.L.W. assisted with adult stem cell derived organoid culture, performed organoid protein extraction, FIS imaging, and analyzed FIS data. S.G. assisted with TEM sample preparation and performed TEM imaging. S.C. performed antimicrobial assays. J.F. assisted with cryo-TEM sample preparation and performed cryo-TEM imaging. B.E.Y. and J.M. collected SANS and USANS data. J.M. performed SANS and USANS data analysis. A.V. performed proteomics data analysis. A.K.N. and K.A.K. wrote the manuscript with input from all authors. N.K., R.D.T., S.A.W., and K.A.K., supervised the work.

## Competing interests

The authors declare no competing interests.
