## [Peer Review File · Nature Communications]

Reviewers' Comments:

Reviewer #1:

Remarks to the Author:

This is an interesting cross length scale study of a biomaterial composed of tryptophan zippers.

The authors demonstrate the Trpzip hydrogels are antimicrobial and self-healing, with tunable viscoelasticity and interesting yield-stress properties. Applications are explored proof of principle by demonstrating their use in cell delivery and bioprinting and disease modelling.

The work is likely to be significant to the field of biomaterials. However, the authors need to clarify how this goes beyond existing literature in the expansive field of peptide/protein based hydrogels.

There is a brief overview in the introduction but more is needed to set the scene for the state of the art in the field, so that the present results can be appreciated. In particular, the authors could help clarify the novelty of the present work by providing a more detailed summary of existing literature on hairpin-based hydrogels and larger peptide based hydrogels, including octa-peptides for which there is significant existing literature, including towards applications in biomedicine. The paper abstract also mentions emergent dynamic properties and it would be useful to clarify to the reader how these differ from existing literature and why this is beneficial.

There may well be novelty but the authors need to clarify how this goes beyond literature such as:

<https://pubs.acs.org/doi/10.1021/acs.biomac.6b01693>

<https://www.sciencedirect.com/science/article/abs/pii/S0003269717303184>

Does the work support the conclusions and claims, or is additional evidence needed?

Yes, if technical comments below are addressed.

Further information could be provided in a number of places to aid clarity, including:

- further details and figures to explain how the SasView program was used to fit the scattering data.
- clarify how the CD data was analysed and whether this was for the hairpin in solution only, or in the hydrogel
- Scales bars needed for Fig SI6
- Scales bars on the microscopy images in the SI (Fig SI8) and further information to validate the approach for sample preparation and staining for this system.
- Clarify number of repeats for data shown in Figure SI9 rheology
- Improve image quality Figure SI10

If the technical details above are addressed and if clarification can be provided as to the novelty of work, then the study could be suitable for publication.

Reviewer #2:

Remarks to the Author:

The authors of "Hierarchical assembly of tryptophan zippers into self-healing bioactive hydrogels" develop a hydrogel based on self-assembling peptides, that fold through Trp-zippers and interact with each other to form nanofibers. They characterize the self assembly and the gel mechanical properties, show fibroblasts can survive encapsulation, and propose applications as an injectable or bioprintable scaffold, as well as an organoid culture scaffold.

I was excited to see a gel design that is rather uncommon and looks promising in theory (even

though there has been already at least a paper exploring very similar Trp-zippers peptide hydrogels published two years ago, that the authors fail to mention, which is a major omission). I was also impressed by how well the stress relaxation profile of Matrigel is recapitulated by this material, and by the range of stiffnesses that can be achieved, as well as the ease with which the gel yields under stress given its stiffness. As such, I think the work is promising. On the other hand, I found the applications lacking. The authors claim that the gel is interesting for in vivo use, can be used for bioprinting, and can be used for intestinal organoid cultures, but there is respectively no data, almost no data / no convincing data, and data that proves the opposite of what the authors would like to claim, for these three applications. I'd recommend either major revisions to fix the issues with applications and adjust the claims, or resubmission at a later timepoint when applications are better under control. I also have quite a number of smaller concerns.

Here would be my detailed comments and recommendations:

"Trpzip sequences have been shown to assemble into nanofibers over the course of several weeks. However, the Trpzip peptide motif has not yet been used to form hydrogels."

There is possibly a big oversight here, or at the very least some detailed explanations critically missing, because Chen et al. 2021 (<https://doi.org/10.1021/acs.jmedchem.1c02081>) claim to obtain various tryptophan-zipper nanofibril-forming peptide hydrogels formulations. I think they should be cited, the claims of novelty should be reconsidered, and whenever possible the properties obtained here should be compared to what was previously done. To start with, they form their gels by incubating for 24h, which already contradicts the claim that "Trpzip sequences have been shown to assemble into nanofibers over the course of several weeks", and brings it down to less than a day. Which also happens to be the same as this study. They also provided CD spectra, tube inversion assays and EM pictures fairly similar to the characterizations done in the first part of this paper. They even showed the Trp were not essential for self assembly to beta hairpins and could be replaced by Leu, which seems relevant to the mechanistic interpretations in this paper. And they also studied the antibacterial properties of these Trpzip hydrogels in great depth, which the authors did here again more superficially, which makes omitting to mention them or compare to them particularly problematic. The authors even go as far as claiming "this is the first example of an extrudable self-healing hydrogel with antimicrobial properties", while a very similar hydrogel had previously been marketed exactly for this application. Granted, Chen et al. did not go in depth on mechanical properties, and use peptides which have the same concept but different sequences. So this paper does bring novelty, but given that Chen use the same self-assembly procedure and get similar nanofibers, one would assume the self-healing properties would be very similar. Since the authors have a peptide synthesizer, it would be

Molecular simulations to optimize the peptide for fibril formation sound like an interesting idea, and the few results shown look exciting. But it seems only 4 single point substitutions on a single residue were considered, and there were no controls to validate the simulation framework (peptides known to fibrils or remain soluble as positive and negative controls). Once the simulations are setup, why not try a large number of variants, not just on the K but also in combinations with other residues? And reporting the results as a dotplot or heatmap in which we get a feeling of which kind of substitution pushes the equilibrium in which direction? The strength of simulations is they can cover a lot of ground rather fast. For only 4 peptide candidates and no controls, setting up the simulation seems like a waste of time, synthesizing them (a 1 to 3 day job) and trying them empirically seems like it would be quicker and would yield more reliable data about which one is best at forming hydrogels. Even though it's a shame it happens at the end of the study, it would still be nice to have this overview of the landscape of self assembly for peptides of this family. Could the simulations also predict the temperature and pH dependence of fiber formation? This would strongly increase the relevance of this first part of the study. Another nice output of the simulations is the structure of the fibrils, and it would be nice that the peptide finally selected (further mutated to have a charge at neutral pH) is part of the simulation landscape so that we get a view of the most relevant structure. The close up image on a single fiber in fig. S1B is quite interesting, would be good to have it for the final chosen peptide, and the close-up might be even more zoomed in so we distinguish individual peptides and how they stack. Are the results of the simulation consistent with the physical data and assembly mechanism shown in Fig2?

I was bugged at first by the claim the gels are yielding at 5% strain. This would mean moving laterally the top of a 1 mm thick gel by 50 um would make the gel fall apart, which would mean the gel is not stable at all in the first place. I then understood the authors did not just apply 5% strain, they applied a cyclic 5% strain, but then the frequency is essential and should be part of the reporting in the main / in the figure (I assume it's 1Hz cf methods), or just report the stress. It's rather the stress applied by the high freq strain than the strain itself which is disrupting the gel probably? In other words, the gel can probably withstand a 5% strain just fine (hopefully) at very low frequency or in a non-oscillatory displacement?

"After exposure to 5% strain for 5 minutes, the hydrogels rapidly re-crosslink, returning to the initial stiffness within an hour"

In the world of hydrogels, bioprinting, and cell culture, I'd say 5-10 min gelling/recovery is average, <1min is fast, <1s is very rapid, but 1h is between slow and very slow. Probably best to just remove the "rapidly" qualifier. For surgical applications in a difficult environment (with bleeding or other fluid flow), minutes is already slow/challenging, subsecond gelling is ideal, an hour to set in is almost an eternity. It's good to keep that in mind when qualifying the speeds and extrapolating applications.

This in particular applies to:

"The low yield point and self-healing properties of Trpzip gels indicate the potential for both injectable cell delivery and extrusion bioprinting of cell-laden inks. During cell injection procedures, mechanical damage results in significant loss of viability at clinically relevant injection rates. We reasoned that Trpzip gels may be able to shield cells from the damaging mechanical forces experienced during flow."

"making Trpzip gels an exciting candidate material for in vivo applications"

To make any claim about interesting properties for clinical/in vivo delivery, the authors should make an actual in vivo delivery, and restrict the claim to the system on which they have tested. As far as my expectations go, the gel recovers far too slow for usage in challenging clinical environments, particularly on a bleeding or wet internal tissue, or a tissue continuously moving (e.g. heart) or with fluid flow (brain/spinal cord, gastrointestinal etc). The gel might be almost OK in a subcutaneous injection procedure, as a guess from the mechanical properties, and might work on the surface of the skin where it can get one hour to recover without being diluted, but may yield too easily to remain in place there afterwards. In any case, I think the claim should be either removed or supported with the relevant data, such as lifetime and functional outcome in vivo in a relevant application of choice.

The authors also make claims about uses of these gels in bioprinting applications, but the only data on this topic is Fig 4H, which is just a schematic of volumetric printing and a very blurry vague line in a low resolution picture, plus dot/line sizes. To claim bioprintability, one would need really way more data, at least demonstrating that complex shapes can be fabricated while retaining cell viability and the complex structures obtained are accurate in shape and sustain themselves, ideally demonstrating some application that pushes the boundaries of what could be done before, demonstrating some interesting property vs a state of the art standard.

I also only see data up to day 5 or 7: this is very short. Is it because the gels fall apart when attempting longer cultures? It is extremely important to estimate and report how long the gels are stable in culture, most cell and organoid cultures, or engineered tissues, require much longer culture durations than a week to mature, checking stability after 3-4 weeks is quite standard, and more is a bonus. Very short lived gels is often a problem.

On a positive note, the relaxation curve of Trpzip vs Matrigel is very impressive and exciting. It's one of the best matches in synthetic gels I have seen so far. Laminin/Matrigel have a peculiar property that they can self-assemble into very soft gels at the bottom of culture dishes when starting from solutions too dilute to form a gel in bulk. Do the Trpzip peptides exhibit this property? It could be another exciting property, not found in many other synthetic gels and definitely not PEG gels. To be clear, this data is not needed for publication, but may be an easy to get additional selling point for the paper. This would open the possibility to use these gels for suspension cultures, which PEG fibrin hyaluronan cellulose etc can't do.

"circumventing the need for large pH switches common to peptide-based hydrogels"

This is a major improvement over RADA16/puramatrix, for which pH issues are a huge handicap that seldom gets reported. It's probably one of the prime reasons organoids are not very commonly grown in puramatrix. I agree it's a nice improvement.

"Integrating Trpzip-IKVAV (10% w/w) into a Trpzip gel resulted in ~10-fold decrease in stiffness"

"We speculate that diluting Trpzip gels with other peptide variants can serve to further tune the structural and mechanical properties as desired."

In an ideal situation, the adhesion cues added do not significantly alter the mechanical properties, so that these design parameters can be tuned independently, so I wouldn't have tried to sell this as a tuning advantage, quite the opposite. On the upside, well designed adhesion peptides are active at less than 100-200 μ M. Blending them as 10% of the structural peptides is probably not the right approach, blending in just the right amount of adhesion cues might not disturb the gel so badly.

Viabilities in Fig4 simultaneously look very good, and a bit misleading: the green channel is highly oversaturated whereas the red channel is very dim, to the extent that I had to zoom in a lot and stare for a while to distinguish the red dots of the dead cells or the doubly stained cells, present in 4G. It's normal that there are a few dead cells, the red and green channels should be shown with a fair color balance. In a high impact journal, I would also expect fibroblast viability, which is really the lowest hanging fruit, to be a detail or supplementary, whereas the main focus should be on novel more interesting cellular behavior in engineered constructs/models.

"Taken together, these data indicate Trpzip hydrogels fosters robust cell attachment, spreading and elongation without the need for cell adhesion cues. We note other synthetic peptide-based hydrogels lack this inherent bio-adhesivity, suggesting Trpzip gels may prove an optimal 3D cell culture material."

Other synthetic hydrogels usually try to separate adhesion from structural properties on purpose, because this enables much more flexibility in the design, e.g. to study adhesion or let some cells and not others adhere or to use the gel both as a barrier or a scaffold depending on the applications. It is also widely known that positively charged polymers are non-specifically sticky to cells (the latter being negatively charged on their surface). This is commonly used in the form of cell culture flask or coverslip coating for cell culture, with poly-L-Lysine, poly-D-lysine, polyornithine, polyethylene imine or other polycation to promote non-specific adhesion, and has been a standard for a number of decades. This peptide being positively charged, it's not a surprised it is therefore behaving like other polycations and supporting cell adhesion and spreading. Adding some pLL peptides in other synthetic gels enables to afford this property on demand. So I would definitely not list it as an advantage or a surprise or a particularly interesting property, just a property. It indeed makes the gel a bit more culture-ready at the expense of tunability. The authors could probably make a negatively charged variant of the same gel to have an inert backbone where adhesion can be studied as in other synthetic gels, and I think this separation of adhesion from mechanics would be preferable. Producing the gel with some click handles like hydrazides, vinyl sulfones, or enzyme ligands, as is done for all the widespread defined matrices, would give the most flexibility to functionalize on demand and would be the preferred way.

"Recent work has demonstrated the importance of laminin protein in promoting the growth of intestinal organoids in synthetic cultures 27. Therefore, we employed a Trpzip variant with the laminin-derived IKVAV peptide at the N-terminus."

There were multiple demonstrating the role of laminin before ref 27, it's not that recent. To my knowledge, the labs doing early work on defined matrices for organoids all tried IKVAV peptides very early on, and found no activity on intestinal organoids. IKVAV was discovered as promoting neurite outgrowth when used as a coating for 2D cultures. It was later found that IKVAV self-assembles into fibrils, as an amyloid peptide which is not rare, and with fiber morphology similar to the gels presented here, forming a very soft layer on top of the coated flasks. The receptor was then found to be APP rather than a laminin receptor (which would instead be e.g. laminin alpha 6 beta 1/4). And the last nail in the coffin of IKVAV as a laminin-mimetic peptide was the discovery that the all D mirror image of the L-peptide had identical biological activity, which is usually considered as the ultimate proof that an effect is physical rather than specific biological

recognition. A key reference for this is Nomizu et al. 1992 ([https://doi.org/10.1016/S0021-9258\(19\)49686-5](https://doi.org/10.1016/S0021-9258(19)49686-5)), but there was a series of papers in the 80s-90s on the topic. Since the authors here already work with a peptide which forms fibers, IKVAV sounds like a complicated and expensive way to change mechanical properties, as well as misleading readers into thinking there is a laminin signal involved when there is not. I know there is quite some recent literature unaware of these facts, that came back to calling IKVAV a laminin mimetic peptide, which can be very confusing unfortunately.

The authors nevertheless had the right intuition that the early labs working on defined gels for organoid cultures found laminin to be key in differentiation conditions. For stem cell expansion, laminin is less essential, so it's best to demonstrate matrix functionality during differentiation, this is the step that needs the engineering. I'd recommend incorporating the real full laminin protein into the gels (which is normally isolated from matrigel, from all major suppliers, unfortunately, which makes all these matrigel replacing gels not entirely independent from matrigel). It's currently the only solution that really works to recapitulate laminin signaling, to my knowledge.

Probably as a consequence of this, the organoid cultures are not working well in my opinion. Failure to polarize indicates the organoids do not recognize the gel as a basement membrane, which should be the job that the synthetic matrix fulfills. In the fluo pictures, we indeed see that the organoids polarize well in Matrigel but fail to polarize in the Trpzip gel, which should have been a signal to the authors that they need to optimize their gel further before attempting publication. In the brightfield images, the organoids look correct in matrigel (light grey color typical of healthy organoids) but look like dying in Trpzip and suspension culture, with a large black mass characteristic of dying cells/debris, and no clear polarized epithelial structure (which would have been a key property expected of intestinal organoids). Suspension cultures should have been possible, they should look like matrigel in a successful situation. It's critical to include 1-2% matrigel in solution in cold medium while seeding the cells. This matrigel re-aggregates around the organoids in suspension and enables them to grow as they do in matrigel. Was this omitted in this study?

Indeed there are some studies with suspension cultures with an inside-out polarity, but of note, their organoids are first generated with matrigel and then flipped, and they do conserve a proper epithelial structure, and look like a clean light-grey-in-brightfield live normal epithelium, rather than a black mass.

I wouldn't try to interpret much the circularity or the presence of a couple more CHGA+ cells in Trpzip gel: circularity is misleading to judge hSI organoids differentiation, nice crypt structures are more typical of mSI organoids so far. The CHGA+ cell number is low, and the difference here appears to be within error bars. The authors found no CHGA+ cell in matrigel, but other authors commonly find these cells, so more likely just a lack of statistics or slightly imperfect culture conditions. It's hard to comment on the transcriptomics until the cultures look like correct organoids, but I see HSPB1 among the top upregulated genes, quite typical of cell stress. And a lot of cytoskeleton related genes in the enriched pathways, which is to be expected if the epithelium structure is not recapitulated (the cells end up in a very unnatural shape and missing their normal adhesions, which would disturb everything cytoskeleton related). CXCR4, Thrombin, spliceosome, and xenobiotic metabolism are all pathways likely to be enriched in response to stress as well.

Given the impressive mechanical properties of Trpzip gels compared to matrigel, I'd encourage the authors to try mouse small intestinal organoids (which are much better understood than the human ones and have more mature culture conditions, and readily form crypts with well organized cells of all types) in their gel in the presence of laminin, while including controls without laminin (negative) and in matrigel (positive), and in differentiation conditions (ENR medium). I think the gel has great potential still needs some tuning and might not have been brought to the application that would emphasize its advantages the most.

A few smaller last remarks:

"revealed the slope in the high Q region was -2 slope"
No need to repeat the word slope.

"dynamic functional properties inherent to natural systems remains an elusive goal"

In the last years there were various system recapitulating the dynamic properties of natural matrices, and matrigel in particular, some quite successfully, so I wouldn't call that an elusive goal. Rather something like an active research domain, a field which has attracted strong interest in recent years, or something like that.

"Peptide amphiphiles are another class of gelators, typically formed from longer peptide (16 residues) with alternating charged amino acids that self-assemble through electrostatic interactions"

There seems to be a small mix up between two things here, might be that the authors are thinking of RADA16 which has both alternating hydrophilic and hydrophobic residues and positive/negative charges. Amphiphiles would be defined as having both hydrophilic and hydrophobic residues/domains (hydrophobic residues, or more typically a lipid tail), and not necessarily any charge interactions. Charge interactions, especially in switzerionic peptides, are another important mechanism indeed, but distinct. And the popular RADA16/puramatrix combines both.

The rheology data looks nice, but I struggle a bit to read quantitative values due to the way it's presented: could we get grid-lines at least on the log-scaled y axis, so we could read stiffness values? Since the datapoints are more or less a continuum, a line instead of these large markers would enable to visualize values more accurately. And error bars (which are needed on all graphs not just 3 out of 8) as an area just like in Fig3F on all panels would be perfect.

The pictures in Fig. 4H and 4I are very low quality or resolution or both, it's hard to distinguish anything in them.

Reviewer #3:

Remarks to the Author:

The manuscript by Nguyen et al reports a tryptophan zipper (Trpzip)-based peptide hydrogel with interesting mechanical and biological properties. The hydrogel exhibits tunable modulus, low yield stress, and self-healing properties, which provide a means for syringe extrusion. In addition, the Trpzip hydrogels is antimicrobial and can be used for propagation of human intestinal organoids. The latter is particularly interesting. The manuscript is well organized. I would suggest acceptance of the manuscript for publication after the authors have corrected the following minor issues.

1. In line 35 page 8, the authors claimed that "this is the first example of an extrudable self-healing hydrogel with antimicrobial properties". However, as far as I know, there are many extrudable self-healing hydrogels with antimicrobial properties that have been reported previously. (e.g. *Biomacromolecules* 2017, 18, 3514-3523; *Biomacromolecules* 2019 20 (5), 1889-1898; *ACS Nano* 2022 16 (5), 7636-7650).

2. Fig. S3 should be cited and discussed in the main text.

3. Fig. S4A shows the frequency distribution of average fiber diameter for Trpzip-QV nanofibers. Please state the pH value with which the sample was prepared.

Reviewer #1:

This is an interesting cross length scale study of a biomaterial composed of tryptophan zippers.

The authors demonstrate the Trpzip hydrogels are antimicrobial and self-healing, with tunable viscoelasticity and interesting yield-stress properties. Applications are explored proof of principle by demonstrating their use in cell delivery and bioprinting and disease modelling.

The work is likely to be significant to the field of biomaterials. However, the authors need to clarify how this goes beyond existing literature in the expansive field of peptide/protein based hydrogels.

We appreciate the positive assessment of our work. In the revised manuscript we have clarified the significance and innovation of the work in the context of the peptide/protein hydrogel field.

There is a brief overview in the introduction but more is needed to set the scene for the state of the art in the field, so that the present results can be appreciated. In particular, the authors could help clarify the novelty of the present work by providing a more detailed summary of existing literature on hairpin-based hydrogels and larger peptide based hydrogels, including octa-peptides for which there is significant existing literature, including towards applications in biomedicine. The paper abstract also mentions emergent dynamic properties and it would be useful to clarify to the reader how these differ from existing literature and why this is beneficial.

There may well be novelty but the authors need to clarify how this goes beyond literature such as:

<https://pubs.acs.org/doi/10.1021/acs.biomac.6b01693>

<https://www.sciencedirect.com/science/article/abs/pii/S0003269717303184>

We have revised the introduction and discussion points to clarify the difference between our material and those presented in earlier studies. The tryptophan zipper sequence we designed is a novel 12-mer peptide that shows numerous new traits that are advantageous compared to previous reports. First, the hydrogels formed by octapeptides (Gao et al, Biomacromolecules, 2017) require pH 3 due to the charge-based assembly of their octapeptide sequences which is a shortcoming that Trpzip gels do not share. We acknowledge Trpzip peptides assemble in a similar fashion to MAX peptides (e.g., Worthington et al, Analytical Biochemistry, 2017) through a beta hairpin secondary structure.

However, the key advance presented by Trpzip peptides is their ability to form hairpins at almost half the peptide length compared to MAX peptides, and without the use of beta turn mimicking units (i.e., d-proline rich sequences). To our knowledge, there has yet to be beta hairpin gelator sequences reported that are not based on the MAX peptides. We believe our findings of a new structural motif that can drive both beta hairpin formation and hydrogelation at neutral pH will support numerous new *in vitro* and *in vivo* hydrogel applications.

We have expanded on these points in the introduction as follows:

“A relatively unexplored assembly motif is the tryptophan zipper (‘Trpzip’). This motif is characterized by four cross-strand tryptophan residues that interlock via the indole rings, folding the peptide into a beta hairpin conformation⁶. As a result of this highly stabilizing ‘zipper’ effect, beta hairpins can be formed from Trpzip peptides that are as short as twelve amino acids. This is considerably shorter than previously reported beta hairpin hydrogelators, such as MAX peptides, which rely on the tetrapeptide (-V^DPPT-) unit and lengthier sequences (>20 amino acids) to maintain a folded hairpin structure¹¹. Previous

work has shown the promise of tryptophan as a building block for self-assembled hydrogels, however these systems still rely on the combined use of other non-natural hydrophobic moieties (e.g., benzene or naphthyl residues)^{12,13} or conventional peptide gelator motifs (e.g., diphenylalanine) to help drive gelation¹⁴."

Further information could be provided in a number of places to aid clarity, including:

- further details and figures to explain how the SasView program was used to fit the scattering data.
- clarify how the CD data was analysed and whether this was for the hairpin in solution only, or in the hydrogel
- Scales bars needed for Fig S16
- Scales bars on the microscopy images in the SI (Fig S18) and further information to validate the approach for sample preparation and staining for this system.
- Clarify number of repeats for data shown in Figure S19 rheology
- Improve image quality Figure S110

We appreciate the careful reading of our manuscript. We have resolved these issues in the revision.

Reviewer #2

The authors of "Hierarchical assembly of tryptophan zippers into self-healing bioactive hydrogels" develop a hydrogel based on self-assembling peptides, that fold through Trp-zippers and interact with each other to form nanofibers. They characterize the self assembly and the gel mechanical properties, show fibroblasts can survive encapsulation, and propose applications as an injectable or bioprintable scaffold, as well as an organoid culture scaffold.

I was excited to see a gel design that is rather uncommon and looks promising in theory (even though there has been already at least a paper exploring very similar Trp-zippers peptide hydrogels published two years ago, that the authors fail to mention, which is a major omission). I was also impressed by how well the stress relaxation profile of Matrigel is recapitulated by this material, and by the range of stiffnesses that can be achieved, as well as the ease with which the gel yields under stress given its stiffness. As such, I think the work is promising. On the other hand, I found the applications lacking. The authors claim that the gel is interesting for in vivo use, can be used for bioprinting, and can be used for intestinal organoid cultures, but there is respectively no data, almost no data / no convincing data, and data that proves the opposite of what the authors would like to claim, for these three applications. I'd recommend either major revisions to fix the issues with applications and adjust the claims, or resubmission at a later timepoint when applications are better under control. I also have quite a number of smaller concerns.

We are glad that the reviewer recognises the promise of our material and apologize if we missed any important papers. In the revised manuscript we have added considerable background to emphasize the novelty and transformational potential of Trpzip hydrogels compared to these earlier studies. In addition, we have performed multiple experiments to further support our claims, while modifying the language to only report attributes that are fully supported by experimental results.

Here would be my detailed comments and recommendations:

"Trpzip sequences have been shown to assemble into nanofibers over the course of several weeks. However, the Trpzip peptide motif has not yet been used to form hydrogels." There is possibly a big oversight here, or at the very least some detailed explanations critically missing, because Chen et al. 2021 (<https://doi.org/10.1021/acs.jmedchem.1c02081>) claim to obtain various tryptophan-zipper nanofibril-forming peptide hydrogels formulations. I think they should be cited, the claims of novelty should be reconsidered, and whenever possible the properties obtained here should be compared to what was previously done. To start with, they form their gels by incubating for 24h, which already contradicts the claim that "Trpzip sequences have been shown to assemble into nanofibers over the course of several weeks", and brings it down to less than a day. Which also happens to be the same as this study. They also provided CD spectra, tube inversion assays and EM pictures fairly similar to the characterizations done in the first part of this paper. They even showed the Trp were not essential for self assembly to beta hairpins and could be replaced by Leu, which seems relevant to the mechanistic interpretations in this paper. And they also studied the antibacterial properties of these Trpzip hydrogels in great depth, which the authors did here again more superficially, which makes omitting to mention them or compare to them particularly problematic. The authors even go as far as claiming "this is the first example of an extrudable self-healing hydrogel with antimicrobial properties", while a very similar hydrogel had previously been marketed exactly for this application. Granted, Chen et al. did not go in depth on mechanical properties, and use peptides which have the same concept but different sequences. So this paper does bring novelty, but given that Chen use the same self-assembly procedure and get similar nanofibers, one would assume the self-healing properties would be very similar. Since the authors have a peptide synthesizer, it would be

We apologize for our failure to cite the study by Chen et al. However, the peptides explored in their work form beta hairpin hydrogels based on assembly motifs other than the tryptophan zipper. We acknowledge the authors incorporated Trp-Trp pairs into a subset of their peptides as a possible way to help stabilise their beta hairpin structure, which is similar to our rationale for designing Trpzip motif variants. However, the central motif in this study (-FFX^DPGXII-), is based on the common diphenylalanine (FF) gelator peptide which is different than our base design.

Moreover, the authors indicate use of a ^DPG turn unit to promote beta hairpin folding (common sequence for initiating beta turns), which is different to assembly of Trpzip-based hairpin structures through crosswise Trp residues. Indeed, the authors show that Trp is not essential for self-assembly of their peptides, which contrasts with our system which requires cross strand Trp residues. Additionally, we note that the antimicrobial properties of their peptides stem from multiple arginine residues instead of the high Trp content in our materials. We acknowledge more work is required to understand the mechanism of antimicrobial activity in Trpzip-based hydrogels, but we believe this is outside of the scope of the current manuscript. Overall, we hold that the core self-assembly mechanism that we present is different and underlies the hierarchical assembly and corresponding extensive properties reported in our paper. We have modified our language to indicate these points in the revised manuscript.

Molecular simulations to optimize the peptide for fibril formation sound like an interesting idea, and the few results shown look exciting. But it seems only 4 single point substitutions on a single residue were considered, and there were no controls to validate the simulation framework (peptides known to fibrils or remain soluble as positive and negative controls). Once the simulations are setup, why not try a large number of variants, not just on the K but also in combinations with other residues? And reporting the results as a dotplot or heatmap in which we

get a feeling of which kind of substitution pushes the equilibrium in which direction? The strength of simulations is they can cover a lot of ground rather fast. For only 4 peptide candidates and no controls, setting up the simulation seems like a waste of time, synthesizing them (a 1 to 3 day job) and trying them empirically seems like it would be quicker and would yield more reliable data about which one is best at forming hydrogels. Even though it's a shame it happens at the end of the study, it would still be nice to have this overview of the landscape of self assembly for peptides of this family. Could the simulations also predict the temperature and pH dependence of fiber formation? This would strongly increase the relevance of this first part of the study. Another nice output of the simulations is the structure of the fibrils, and it would be nice that the peptide finally selected (further mutated to have a charge at neutral pH) is part of the simulation landscape so that we get a view of the most relevant structure. The close up image on a single fiber in fig. S1B is quite interesting, would be good to have it for the final chosen peptide, and the close-up might be even more zoomed in so we distinguish individual peptides and how they stack. Are the results of the simulation consistent with the physical data and assembly mechanism shown in Fig2?

We appreciate the reviewer's guidance on this point. In the revised manuscript we have compiled simulation data for all twenty variants and presented it as a heatmap to show the effect of amino acid substitution on aggregate morphology (**Rebuttal Figure 1a and Figure 1e**). In addition, we have run the same simulation framework on control peptide sequences as requested (**Rebuttal Figure 1b–d**). The tripeptides KFD and GGG were selected as the positive and negative control, respectively. These were chosen based on the work of Frederix *et al.* (Nature Chemistry, 2014) who similarly used the Martini coarse-grain force field to model aggregation of all possible tripeptide sequences and experimentally validated KFD as a highly efficient gelator, and GGG as a non-aggregating peptide. We have also compared the final variant through simulations as requested (**Rebuttal Figure 1e–f**), although we note the results do not suggest highly fibrillar aggregates are formed. We speculate this discrepancy could be due to the difference in charge state of Trpzip-QV compared to Trpzip-V, with increasing electrostatic effects more likely to dispel aggregation. We also agree with the reviewer that further atomistic resolution simulations could provide insight into the assembly mechanism; however, we believe the coarse-grain data (although backmapped to all-atom) is still not refined enough to reliably extrapolate the precise organisation of individual peptides within the nanofiber. We hope to explore this further in future all-atom studies, as well as investigating temperature and pH dependence on simulated fiber formation, as suggested by the reviewer.

Rebuttal Figure 1. MD simulations of peptide aggregation and nanofiber formation using the Martini force field. A) Moments of inertia along each axis for the largest cluster of peptides of Trpzip variants substituted with all twenty canonical amino acids. B) Final frame of 100 μs simulation of the tripeptide KFD, used as a positive control for peptide aggregation. C) Final frame of 100 μs simulation of the tripeptide GGG, used as a negative control for peptide aggregation. D) Moments of inertia along each axis for the largest cluster of peptides for KFD and GGG. E) Final frame of 100 μs simulation of Trpzip-QV. F) Moments of inertia along each axis for the largest cluster of peptides comparing the original Trpzip1 sequence, and variants Trpzip-V and Trpzip-QV.

I was bugged at first by the claim the gels are yielding at 5% strain. This would mean moving laterally the top of a 1 mm thick gel by 50 μm would make the gel fall apart, which would mean the gel is not stable at all in the first place. I then understood the authors did not just apply 5% strain, they applied a cyclic 5% strain, but then the frequency is essential and should be part of the reporting in the main / in the figure (I assume it's 1 Hz cf methods), or just report the stress. It's rather the stress applied by the high freq strain than the strain itself which is disrupting the gel probably? In other words, the gel can probably withstand a 5% strain just fine (hopefully) at very low frequency or in a non-oscillatory displacement?

We agree that in oscillatory measurements, the stress is likely what leads to the gels yielding. We have updated the manuscript to mention the yield stress instead of the yield strain to improve clarity, and we have updated the caption with the frequency used for the strain amplitude sweeps:

“Surprisingly, the yield point of Trpzip gels occurs at a shear stress of approximately 75 Pa (Fig. 3C).” and “A strain sweep performed at 1 Hz of a Trpzip-QV hydrogel (1% w/v, DMEM, pH 7).

Dotted pink line indicates yield-point. Data is shown as mean±s.d. (shaded area) from n = 2 independently prepared gels., respectively.

We also performed a frequency amplitude sweep test to verify that the frequency used for our measurements for oscillatory tests (1 Hz) was well within the linear viscoelastic region (**Figure S9**).

"After exposure to 5% strain for 5 minutes, the hydrogels rapidly re-crosslink, returning to the initial stiffness within an hour"

In the world of hydrogels, bioprinting, and cell culture, I'd say 5-10 min gelling/recovery is average, <1min is fast, <1s is very rapid, but 1h is between slow and very slow. Probably best to just remove the "rapidly" qualifier. For surgical applications in a difficult environment (with bleeding or other fluid flow), minutes is already slow/challenging, subsecond gelling is ideal, an hour to set in is almost an eternity. It's good to keep that in mind when qualifying the speeds and extrapolating applications.

We thank the reviewer for bringing up this important point and agree that "rapid" is not appropriate for 1 hour in the broader context of modern hydrogels. We conducted several experiments where we evaluated alternative gel casting protocols to try and improve gelation time. We are delighted to report that we achieved fast hydrogelation (within seconds) by first reconstituting the peptide in deionized water followed by adding cell culture media to a final concentration of 0.5% (w/v) (**Rebuttal Figure 2** and **Figure S10**). This is superior to other peptide hydrogelators that invariably require high and low pH triggers, such as RADA16/Pura Matrix. Nevertheless, since the word rapid is subjective, we have removed it from our description as suggested by the reviewer.

While this new gel casting method still results in the same thixotropic self-healing behaviour, we believe this near-instantaneous method of triggering gelation from a peptide solution in water increases the suitability for practical applications where timing and application simplicity are critical.

Rebuttal Figure 2. Modulus over time for modified Trpzip casting method via ionically triggered gelation allows for comparable mechanics at half the peptide concentration. Data are represented as mean±s.d. (shaded area) from n = 3 independently prepared hydrogels.

This in particular applies to:

"The low yield point and self-healing properties of Trpzip gels indicate the potential for both

injectable cell delivery and extrusion bioprinting of cell-laden inks. During cell injection procedures, mechanical damage results in significant loss of viability at clinically relevant injection rates. We reasoned that Trpzip gels may be able to shield cells from the damaging mechanical forces experienced during flow."

"making Trpzip gels an exciting candidate material for in vivo applications"

To make any claim about interesting properties for clinical/in vivo delivery, the authors should make an actual in vivo delivery, and restrict the claim to the system on which they have tested. As far as my expectations go, the gel recovers far too slow for usage in challenging clinical environments, particularly on a bleeding or wet internal tissue, or a tissue continuously moving (e.g. heart) or with fluid flow (brain/spinal cord, gastrointestinal etc). The gel might be almost OK in a subcutaneous injection procedure, as a guess from the mechanical properties, and might work on the surface of the skin where it can get one hour to recover without being diluted, but may yield too easily to remain in place there afterwards. In any case, I think the claim should be either removed or supported with the relevant data, such as lifetime and functional outcome in vivo in a relevant application of choice.

We agree with the reviewer that to claim clinical utility there needs to be supporting evidence. Our intention was to present new directions that we believe hold potential for Trpzip materials, but which are outside of the scope of the current manuscript. We have modified our language to alleviate any confusion and removed claims of *in vivo* applications as follows:

"The low yield point and self-healing properties of Trpzip gels opens the potential for syringe delivery. In liquid cell suspensions, mechanical damage results in significant loss of viability²⁰. We reasoned that Trpzip gels may be able to shield cells from the damaging mechanical forces experienced during flow."

The authors also make claims about uses of these gels in bioprinting applications, but the only data on this topic is Fig 4H, which is just a schematic of volumetric printing and a very blurry vague line in a low resolution picture, plus dot/line sizes. To claim bioprintability, one would need really way more data, at least demonstrating that complex shapes can be fabricated while retaining cell viability and the complex structures obtained are accurate in shape and sustain themselves, ideally demonstrating some application that pushes the boundaries of what could be done before, demonstrating some interesting property vs a state of the art standard.

As above, here we were merely pointing to interesting properties of the hydrogel that might lead to new applications. We have carefully reworded this section to ensure there are no claims not supported by data as follows:

"The yield-stress fluid properties provide scope for Trpzip gels to serve as a support bed for deposition of material; that is, the ability to fluidize under shear and self-heal around a deposited material. Here, we extruded a high-density of fibroblast cells into droplets and lines within a Trpzip gel support (Fig. 4H; Fig. S13). Dot diameters averaged $650 \pm 80 \mu\text{m}$ ($n=8$), and line diameters averaging $350 \pm 50 \mu\text{m}$ ($n=4$; 14% from theoretical line width). Immunofluorescent imaging of the embedded solutions show they consist of tightly packed cells (Fig. 4H), demonstrating the Trpzip gel supports syringe deposition of cell suspensions."

I also only see data up to day 5 or 7: this is very short. Is it because the gels fall apart when attempting longer cultures? It is extremely important to estimate and report how long the gels are stable in culture, most cell and organoid cultures, or engineered tissues, require much

longer culture durations than a week to mature, checking stability after 3-4 weeks is quite standard, and more is a bonus. Very short lived gels is often a problem.

We agree with the reviewer that long term culture is important for many applications. In the original manuscript we did include data showing fibroblast cells in culture for 2 weeks (**Figure 4F; Fig S14**). Nevertheless, we have since performed extended cultures of fibroblast cells (**Rebuttal Figure 3a; Fig. S13**), as well as a common myoblast progenitor C2C12 cells, for one month as a demonstration of the ability of our material to support long-term tissue culture. The fibroblast cells continued to spread and proliferate, and we found the hydrogel material stayed intact to a far greater extent than Matrigel (**Rebuttal Figure 3b**). In contrast, Matrigel was largely remodelled over this time leading to considerable loss of overall mass and structure (**Rebuttal Figure 3c**). Further, compared to glass 2D cultures, the C2C12 cells formed dense and aligned structures that persisted over long distances within the Trpzip hydrogel and remained a stable solid material for 1 month (**Rebuttal Figure 3d**).

Rebuttal Figure 3. Culture of mammalian cells in Trpzip gels for one month. A) Brightfield images of fibroblast (HFF-1s) cells cultured in Matrigel and Trpzip gels, showing state of respective gels over 30 days. B) Photographs of 96-well culture dish containing fibroblast and myoblast cells cultures, showing hydrogel state after 30 days. Red arrow in Matrigel sample shows section of Matrigel that has detached from glass by Day 30. Bottom photograph shows cell laden Trpzip gel after 30 days. Scale is 50 mm. C) Immunofluorescence images of

fibroblasts cultured in 3D in Matrigel and Trpzip gels, stained for F-actin (green) and nuclei (blue). Scale is 50 μm . D) Immunofluorescence images of myoblasts cultured in 2D on glass, and 3D in Trpzip gels, stained for F-actin (green) and nuclei (blue). Scale is 50 μm .

On a positive note, the relaxation curve of Trpzip vs Matrigel is very impressive and exciting. It's one of the best matches in synthetic gels I have seen so far. Laminin/Matrigel have a peculiar property that they can self-assemble into very soft gels at the bottom of culture dishes when starting from solutions too dilute to form a gel in bulk. Do the Trpzip peptides exhibit this property? It could be another exciting property, not found in many other synthetic gels and definitely not PEG gels. To be clear, this data is not needed for publication, but may be an easy to get additional selling point for the paper. This would open the possibility to use these gels for suspension cultures, which PEG fibrin hyaluronan cellulose etc can't do.

We appreciate the reviewer bringing this interesting idea to our attention. We performed the experiment as suggested and found Trpzip solutions at concentration below the gelation point (0.1 wt%) did indeed form a soft layer of gel at the bottom of a culture dish similar to Matrigel. (**Rebuttal Figure 4a**). We confirmed with rheological measurements that the peptide forms a very weak 10 Pa gel at this concentration (**Rebuttal Figure 4b**).

Rebuttal Figure 4. Formation of Trpzip gel following overnight incubation of a 0.1 wt% solution of Trpzip peptide in DMEM/F12 at 37 °C. A) Photographs of Matrigel and Trpzip gels after removal of residual liquid media in culture dish. Surface of Matrigel has been scratched to help visualisation of gel layer. B) Time sweep of 0.1 wt% Trpzip gel in DMEM/F12 at 37 °C.

"circumventing the need for large pH switches common to peptide-based hydrogels"

This is a major improvement over RADA16/puramatrix, for which pH issues are a huge handicap that seldom gets reported. It's probably one of the prime reasons organoids are not very commonly grown in puramatrix. I agree it's a nice improvement.

We thank the reviewer for the positive assessment of this property.

"Integrating Trpzip-IKVAV (10% w/w) into a Trpzip gel resulted in ~10-fold decrease in stiffness"

"We speculate that diluting Trpzip gels with other peptide variants can serve to further tune the structural and mechanical properties as desired."

In an ideal situation, the adhesion cues added do not significantly alter the mechanical properties, so that these design parameters can be tuned independently, so I wouldn't have tried to sell this as a tuning advantage, quite the opposite. On the upside, well designed adhesion peptides are active at less than 100-200 μM . Blending them as 10% of the structural peptides is probably not the right approach, blending in just the right amount of adhesion cues might not disturb the gel so badly.

We understand the reviewer's point. In response, we repeated the measurements and were surprised to see negligible change in mechanics upon addition of the peptide (**Figure 3H** and **Rebuttal Figure 5**). Upon investigation we found that the first Trpzip-IKVAV was synthesised as a different salt from a commercial vendor compared to our in-house peptides (TFA salt versus formate salt), which we speculate led to the destabilisation of the hydrogel. As such, we have replaced this data and removed the discussion about tuning modulus through peptide addition.

Rebuttal Figure 5. Elastic modulus of trpzip with addition of adhesion peptides (200 μ M). Data are represented as mean \pm s.d. (shaded area) from $n = 3$ independently prepared hydrogels.

Viabilities in Fig4 simultaneously look very good, and a bit misleading: the green channel is highly oversaturated whereas the red channel is very dim, to the extent that I had to zoom in a lot and stare for a while to distinguish the red dots of the dead cells or the doubly stained cells, present in 4G. It's normal that there are a few dead cells, the red and green channels should be shown with a fair color balance. In a high impact journal, I would also expect fibroblast viability, which is really the lowest hanging fruit, to be a detail or supplementary, whereas the main focus should be on novel more interesting cellular behavior in engineered constructs/models.

We apologize for any misleading information. The viability image in **Figure 4A** indeed shows no dead cells. We agree that it is normal that there will be a few dead cells present, so we have updated the manuscript with another image from the same experiment that does show some dead cells for a more balanced representation of the data.

Since these images are used to demonstrate viability across numerous different formats with future scope in mind, we believe they are appropriate for a main figure.

"Taken together, these data indicate Trpzip hydrogels fosters robust cell attachment, spreading and elongation without the need for cell adhesion cues. We note other synthetic peptide-based hydrogels lack this inherent bio-adhesivity, suggesting Trpzip gels may prove an optimal 3D cell culture material."

Other synthetic hydrogels usually try to separate adhesion from structural properties on purpose, because this enables much more flexibility in the design, e.g. to study adhesion or let some cells and not others adhere or to use the gel both as a barrier or a scaffold depending on the applications. It is also widely known that positively charged polymers are non-specifically sticky to cells (the latter being negatively charged on their surface). This is commonly used in the form of cell culture flask or coverslip coating for cell culture, with poly-L-Lysine, poly-D-lysine,

polyornithine, polyethylene imine or other polycation to promote non-specific adhesion, and has been a standard for a number of decades. This peptide being positively charged, it's not a surprise it is therefore behaving like other polycations and supporting cell adhesion and spreading. Adding some pLL peptides in other synthetic gels enables to afford this property on demand. So I would definitely not list it as an advantage or a surprise or a particularly interesting property, just a property. It indeed makes the gel a bit more culture-ready at the expense of tunability. The authors could probably make a negatively charged variant of the same gel to have an inert backbone where adhesion can be studied as in other synthetic gels, and I think this separation of adhesion from mechanics would be preferable. Producing the gel with some click handles like hydrazides, vinyl sulfones, or enzyme ligands, as is done for all the widespread defined matrices, would give the most flexibility to functionalize on demand and would be the preferred way.

As requested, we have modified our language appropriately. We agree that adding modular handles to the peptides for further modification will be a great next step to demonstrate versatility. However, we believe this is outside of the scope of the current manuscript.

"Recent work has demonstrated the importance of laminin protein in promoting the growth of intestinal organoids in synthetic cultures 27. Therefore, we employed a Trpzip variant with the laminin-derived IKVAV peptide at the N-terminus."

There were multiple demonstrating the role of laminin before ref 27, it's not that recent. To my knowledge, the labs doing early work on defined matrices for organoids all tried IKVAV peptides very early on, and found no activity on intestinal organoids. IKVAV was discovered as promoting neurite outgrowth when used as a coating for 2D cultures. It was later found that IKVAV self-assembles into fibrils, as an amyloid peptide which is not rare, and with fiber morphology similar to the gels presented here, forming a very soft layer on top of the coated flasks. The receptor was then found to be APP rather than a laminin receptor (which would instead be e.g. laminin alpha 6 beta 1/4). And the last nail in the coffin of IKVAV as a laminin-mimetic peptide was the discovery that the all D mirror image of the L-peptide had identical biological activity, which is usually considered as the ultimate proof that an effect is physical rather than specific biological recognition. A key reference for this is Nomizu et al. 1992 ([https://doi.org/10.1016/S0021-9258\(19\)49686-5](https://doi.org/10.1016/S0021-9258(19)49686-5)), but there was a series of papers in the 80s-90s on the topic. Since the authors here already work with a peptide which forms fibers, IKVAV sounds like a complicated and expensive way to change mechanical properties, as well as misleading readers into thinking there is a laminin signal involved when there is not. I know there is quite some recent literature unaware of these facts, that came back to calling IKVAV a laminin mimetic peptide, which can be very confusing unfortunately.

We thank the reviewer for the guidance and apologize for any confusion regarding the IKVAV peptide. In the revised manuscript we have modified our language to indicate IKVAV is a commonly used adhesive sequence for biomaterials, but not a substitute for full-length laminin.

The authors nevertheless had the right intuition that the early labs working on defined gels for organoid cultures found laminin to be key in differentiation conditions. For stem cell expansion, laminin is less essential, so it's best to demonstrate matrix functionality during differentiation, this is the step that needs the engineering. I'd recommend incorporating the real full laminin protein into the gels (which is normally isolated from matrigel, from all major suppliers, unfortunately, which makes all these matrigel replacing gels not entirely independent from matrigel). It's currently the only solution that really works to recapitulate laminin signaling, to my knowledge.

We are grateful for this excellent suggestion. As anticipated by the reviewer, including full length laminin into Trpzip hydrogels leads to improved basement membrane mimicry with basal polarity and budding. We now demonstrate how we can direct apical-basal polarity through mixing the Trpzip peptide with laminin. We have added this data to the new **Figure 5** and **Rebuttal Figure 6** below, with accompanying text in results and discussion.

Rebuttal Figure 6. Effect of adding laminin to Trpzip hydrogels on organoid growth and polarisation compared to Matrigel. A) Hierarchical clustering analysis of proteomes of organoids grown in Matrigel, Trpzip gels, and Trpzip gels with low and high laminin. B) and C) Forskolin swelling assay of organoids grown in Matrigel, Trpzip gels, and Trpzip gels with low and high laminin. Scale is 200 μm . Organoid viability was confirmed through Calcein AM staining.

Probably as a consequence of this, the organoid cultures are not working well in my opinion. Failure to polarize indicates the organoids do not recognize the gel as a basement membrane, which should be the job that the synthetic matrix fulfills. In the fluo pictures, we indeed see that the organoids polarize well in Matrigel but fail to polarize in the Trpzip gel, which should have been a signal to the authors that they need to optimize their gel further before attempting publication. In the brightfield images, the organoids look correct in matrigel (light grey color typical of healthy organoids) but look like dying in Trpzip and suspension culture, with a large black mass characteristic of dying cells/debris, and no clear polarized epithelial structure (which would have been a key property expected of intestinal organoids). Suspension cultures should have been possible, they should look like matrigel in a successful situation. It's critical to include 1-2% matrigel in solution in cold medium while seeding the cells. This matrigel re-aggregates around the organoids in suspension and enables them to grow as they do in matrigel. Was this omitted in this study?

We repeated these experiments in Trpzip gels with and without laminin, including Calcein AM staining and a forskolin swelling assay to verify polarity. As shown in the new **Figure 5** and **Rebuttal Figure 6**, Trpzip alone leads to viable organoids with a predominantly apical polarity, like those created in suspension culture. The addition of laminin leads to basal-out polarity and increased budding/formation of crypts, similar to Matrigel (**Rebuttal Figure 6b**). Live cell imaging after treatment with the CFTR potentiator Trikafta demonstrates swelling of the organoids embedded in Matrigel or laminin supplemented Trpzip, where swelling indicates

basal-out polarity of organoids with DF508-CFTR mutation (**Rebuttal Figure 6c-d**). In contrast, Trpzip alone or with low laminin content leads to either no change or shrinkage of the embedded organoids. This data provides evidence that the organoids are viable and have an apical out polarity in Trpzip gel, which can be modified by addition of laminin.

The following discussion was added to the revised manuscript:

“Previous work has demonstrated how the presence of laminin protein is important for supporting basal polarity and growth in synthetic hydrogels²⁷. While apical facing organoids hold several advantages as discussed, we next sought to evaluate whether the addition of laminin would influence the polarity and growth characteristics in Trpzip encapsulated organoids. We blended Trpzip hydrogels with varying amounts of laminin to determine if we could mimic the polarity and growth characteristics of organoids grown in Matrigel. We cultured organoids from one participant in Matrigel, pure Trpzip gels, Trpzip gels incorporating 0.4 mg/ml laminin (low), and Trpzip gels incorporating 3 mg/ml laminin (high). Organoids grown in pure Trpzip gels and those containing low laminin maintained a spherical morphology (Fig. S16). In contrast, organoids grown in the higher laminin content Trpzip gels appeared similar in morphology to organoids grown in Matrigel. Global proteomics analysis shows significant similarity between all conditions – pure Trpzip, Trpzip with low and high laminin, and Matrigel (Fig. 5H). Importantly, we observe clustering of Trpzip/Trpzip-low laminin and Matrigel/Trpzip-high laminin, suggesting greater similarity to Matrigel as we increase the laminin content in the matrix (Fig. 5H; Fig. S21A–B).

Since the addition of laminin to Trpzip appears to more closely emulate growth in Matrigel, we performed a forskolin swelling assay to quantitatively assess organoid polarity. As the organoids were derived from patients with the homozygous DF508-CFTR mutation, organoids with a basal-out polarity are expected to swell upon correction and activation of the cystic fibrosis transmembrane conductance regulator (CFTR) channel. Upon activation of the CFTR channel, organoid swelling was observed within 120 minutes with live imaging (Fig. 5J–K; Supplementary Video 2–5). Conversely, organoids grown in Trpzip gels and Trpzip with low laminin exhibited no such swelling and in some organoid shrinkage was observed, confirming Trpzip-grown organoids display apical-out polarity. Interestingly, organoids grown in Trpzip gels containing high laminin display swelling, indicating preservation of a basal-out polarity. This finding substantiates the fact that basal-out polarity can be maintained in organoids grown in Trpzip hydrogels when a sufficient concentration of laminin protein is present. This set of data suggests that control of polarity with Trpzip hydrogels is possible and laminin supplementation can be used to mimic growth in Matrigel.”

Indeed there are some studies with suspension cultures with an inside-out polarity, but of note, their organoids are first generated with matrigel and then flipped, and they do conserve a proper epithelial structure, and look like a clean light-grey-in-brightfield live normal epithelium, rather than a black mass.

As requested, we generated our organoids in Matrigel followed by deposition within Trpzip gels or suspension culture. However, we continue to see the dark appearance which we believe is on account of light scattering within the apical-out organoid as observed by other authors (**Rebuttal Figure 7**; adapted from Figure 1B of Co *et al*, Cell Reports, 2019). Considering our positive Calcein staining, evidence for growth over time and response to FIS treatment, we believe our organoids are viable.

Rebuttal Figure 7. Images from a dissection microscope of BME-embedded enteroids (left) or suspended enteroids (right). Scale bar is 500 μm . (Figure caption taken from cited paper).

I wouldn't try to interpret much the circularity or the presence of a couple more CHGA+ cells in Trpzip gel: circularity is misleading to judge hSI organoids differentiation, nice crypt structures are more typical of mSI organoids so far. The CHGA+ cell number is low, and the difference here appears to be within error bars. The authors found no CHGA+ cell in matrigel, but other authors commonly find these cells, so more likely just a lack of statistics or slightly imperfect culture conditions. It's hard to comment on the transcriptomics until the cultures look like correct organoids, but I see HSPB1 among the top upregulated genes, quite typical of cell stress. And a lot of cytoskeleton related genes in the enriched pathways, which is to be expected if the epithelium structure is not recapitulated (the cells end up in a very unnatural shape and missing their normal adhesions, which would disturb everything cytoskeleton related). CXCR4, Thrombin, spliceosome, and xenobiotic metabolism are all pathways likely to be enriched in response to stress as well.

We appreciate the guidance from the reviewer. In response we have removed the circularity and CHGA immunostaining data and limited our interpretation of the original proteomics results as recommended. We have performed global proteomics of organoids grown in laminin-supplemented Trpzip (**Rebuttal Figure 6e-f**), and have included this in the revised manuscript as follows:

“Global proteomics analysis shows significant similarity between all conditions – pure Trpzip, Trpzip with low and high laminin, and Matrigel (Fig. 5H; Fig. S20A–B). Importantly, we observe clustering of Trpzip/Trpzip-low laminin and Matrigel/Trpzip-high laminin, suggesting greater similarity to Matrigel as we increase the laminin content in the matrix (Fig. 5H; Fig. S20A–B).”

Given the impressive mechanical properties of Trpzip gels compared to matrigel, I'd encourage the authors to try mouse small intestinal organoids (which are much better understood than the human ones and have more mature culture conditions, and readily form crypts with well organized cells of all types) in their gel in the presence of laminin, while including controls without laminin (negative) and in matrigel (positive), and in differentiation conditions (ENR medium). I think the gel has great potential still needs some tuning and might not have been brought to the application that would emphasize its advantages the most.

We appreciate the recommendation, and we will look forward to trying the mouse small intestinal organoids in the future. However, we believe this is beyond the scope of this manuscript.

A few smaller last remarks:

"revealed the slope in the high Q region was -2 slope"

No need to repeat the word slope.

This has been corrected.

"dynamic functional properties inherent to natural systems remains an elusive goal"

In the last years there were various system recapitulating the dynamic properties of natural matrices, and matrigel in particular, some quite successfully, so I wouldn't call that an elusive goal. Rather something like an active research domain, a field which has attracted strong interest in recent years, or something like that.

We have modified this sentence as requested.

"Peptide amphiphiles are another class of gelators, typically formed from longer peptide (16 residues) with alternating charged amino acids that self-assemble through electrostatic interactions"

There seems to be a small mix up between two things here, might be that the authors are thinking of RADA16 which has both alternating hydrophilic and hydrophobic residues and positive/negative charges. Amphiphiles would be defined as having both hydrophilic and hydrophobic residues/domains (hydrophobic residues, or more typically a lipid tail), and not necessarily any charge interactions. Charge interactions, especially in switzerionic peptides, are another important mechanism indeed, but distinct. And the popular RADA16/puramatrix combines both.

We have expanded this section considerably to ensure that all nuances of these different peptide hydrogelators has been clarified.

The rheology data looks nice, but I struggle a bit to read quantitative values due to the way it's presented: could we get grid-lines at least on the log-scaled y axis, so we could read stiffness values? Since the datapoints are more or less a continuum, a line instead of these large markers would enable to visualize values more accurately. And error bars (which are needed on all graphs not just 3 out of 8) as an area just like in Fig3F on all panels would be perfect.

We have added gridlines and error bars as requested.

The pictures in Fig. 4H and 4I are very low quality or resolution or both, it's hard to distinguish anything in them.

We have updated these figures with high resolution images.

Reviewer #3

The manuscript by Nguyen et al reports a tryptophan zipper (Trpzip)-based peptide hydrogel with interesting mechanical and biological properties. The hydrogel exhibits tunable modulus, low yield stress, and self-healing properties, which provide a means for syringe extrusion. In addition, the Trpzip hydrogels is antimicrobial and can be used for propagation of human intestinal organoids. The latter is particularly interesting. The manuscript is well organized. I would suggest acceptance of the manuscript for publication after the authors have corrected the following minor issues.

1. In line 35 page 8, the authors claimed that "this is the first example of an extrudable self-healing hydrogel with antimicrobial properties". However, as far as I know, there are many extrudable self-healing hydrogels with antimicrobial properties that have been reported

previously. (e.g. Biomacromolecules 2017, 18, 3514-3523; Biomacromolecules 2019 20 (5), 1889-1898; ACS Nano 2022 16 (5), 7636-7650).

We appreciate the guidance from the reviewer. We have removed this claim and added these references to our discussion.

“Like other injectable antimicrobial hydrogels^{23,24}, Trpzip-based materials may prove useful as extrudable self-healing antimicrobial hydrogels, with potential for use in medical applications.”

While there are examples of antimicrobial gels with self-healing characteristics, we believe there are numerous advantages with the Trpzip material which we highlight in the revised manuscript.

2. Fig. S3 should be cited and discussed in the main text.
3. Fig. S4A shows the frequency distribution of average fiber diameter for Trpzip-QV nanofibers. Please state the pH value with which the sample was prepared.

We have corrected these errors in the revised manuscript.

Reviewers' Comments:

Reviewer #2:

Remarks to the Author:

The authors did a great job at addressing the concerns I had with the previous version of the manuscript. I think they have solved several major issues which makes the work much more impactful: gelling time, organoid viability and functionality, discussion of the originality compared to previous work in the field, getting interesting info out of their simulations, and being more careful that the claims made don't extrapolate too much from the data shown.

I'd still have a small disagreement about the interpretation of the organoids in the Trp-zipper gels only - to me the calcein positive areas in the updated figure indeed correspond to the parts which were light grey in brightfield, and are only ~20% of the organoid area in the pictures shown, so I'd still consider the organoids to be mostly dying in these gels, even more so with this calcein double-check. Nevertheless, adding laminin resolved the issue and the functionality, and the pictures speak for themselves, so this claim becomes much less important now.

They also appear to be polarized towards both sides (or in other words not really polarized) rather than simply reverse polarized, according to the immunostainings. That is, microvilli point towards both the lumen and the outside, and the cells only adhere to each other, not to the matrix, in Trp-zipper gels not supplemented with laminin. But here again the pictures in the updated main figure are clear enough to make the claims less important. If possible the authors might still make little adjustments on these points to be entirely accurate, but otherwise I think the manuscript is ready for publication.

Congratulations on the interesting work and best wishes.

Reviewer #3:

Remarks to the Author:

As required by the editor, I have assessed the responses of the authors given to the comments of Reviewer #1. The reviewer #1 made a positive overall comment to the manuscript, and raised some technical issues for the authors "to aid clarity". I think these concerns have been properly addressed in the revised manuscript.

In addition, the authors have properly addressed my comments in the revised manuscript. Therefore, I would recommend acceptance of the manuscript for publication in NC.

Reviewer #2:

The authors did a great job at addressing the concerns I had with the previous version of the manuscript. I think they have solved several major issues which makes the work much more impactful: gelling time, organoid viability and functionality, discussion of the originality compared to previous work in the field, getting interesting info out of their simulations, and being more careful that the claims made don't extrapolate too much from the data shown.

We are delighted to hear that our revision has satisfied the reviewer.

I'd still have a small disagreement about the interpretation of the organoids in the Trp-zipper gels only - to me the calcein positive areas in the updated figure indeed correspond to the parts which were light grey in brightfield, and are only ~20% of the organoid area in the pictures shown, so I'd still consider the organoids to be mostly dying in these gels, even more so with this calcein double-check. Nevertheless, adding laminin resolved the issue and the functionality, and the pictures speak for themselves, so this claim becomes much less important now.

We appreciate the reviewer pointing out this issue which could prove misleading to readers. However, we disagree with the interpretation. We believe the lighter calcein staining in the dark regions is merely an effect of decreased light emission from these regions and not due to cell death; this has been reported previously for calcein staining (Al-Abd et al., Cancer Sci, 2008, 99 (2), 423). Indeed, we see this variability across all organoids irrespective of material. This is confirmed by live/dead imaging, where we clearly see heterogeneity in calcein stains but no evidence of correlation with cell death (red stain from propidium iodide; **Rebuttal figure 1**).

Rebuttal Figure 1. Live dead imaging of organoids in Matrigel, trpzip gel, and trpzip including laminin.

There is a small number of dead cells in all conditions, which is expected in complex organoid cultures where morphogenesis requires some degree of apoptosis. We do see a slightly higher degree of cell death in the Trpzip only gels, presumably due to programmed cell death as the

organoids change their polarity, which is reduced as we add laminin protein. This has been discussed in the cited paper by Co et al., Cell Rep. 2019, 26 (9), 2509 (ref 28), where a switch from basal to apical polarity coincides with dead cells being expelled outward. Further evidence for the Trpzip-grown organoid viability arises from our immunofluorescence study, where cells are uniformly expressing intestinal markers throughout (Figure 5D,E). Nevertheless, to avoid any further confusion, we have added the live/dead data as a new panel to **Figure S19C** with the corresponding text added to the associated discussion:

“Live/dead imaging indicates a slight increase in cell death for the Trpzip-only conditions, which we attribute to normal apoptotic signalling during morphogenesis as observed previously with apical-out organoid protocols²⁸(Figure S19C). As expected, the addition of laminin increases viability in the cultures.”

They also appear to be polarized towards both sides (or in other words not really polarized) rather than simply reverse polarized, according to the immunostainings. That is, microvilli point towards both the lumen and the outside, and the cells only adhere to each other, not to the matrix, in Trp-zipper gels not supplemented with laminin. But here again the pictures in the updated main figure are clear enough to make the claims less important. If possible the authors might still make little adjustments on these points to be entirely accurate, but otherwise I think the manuscript is ready for publication.

We appreciate the reviewer raising this point. Early organoids embedded in Trpzip show apical-out polarity. However, once they grow larger than 100 μm in diameter, they can develop an internal lumen as shown in Figure 5B. While this does not change our conclusions or the potential utility of these organoids for use in “apical-out” applications, it is important to make this clear to the readers. As such we have added the following sentence:

“Once the Trpzip embedded organoids grow larger (>100 μm), there is evidence for both outward and inward polarization with an internal lumen. This characteristic will not impede studies of apical facing organoid behavior.”

We have also changed the section title wording from “polarity reversal” to “polarity changes” to correspond with the observations.

Reviewer #3

As required by the editor, I have assessed the responses of the authors given to the comments of Reviewer #1. The reviewer #1 made a positive overall comment to the manuscript, and raised some technical issues for the authors “to aid clarity”. I think these concerns have been properly addressed in the revised manuscript.

In addition, the authors have properly addressed my comments in the revised manuscript. Therefore, I would recommend acceptance of the manuscript for publication in NC.

We appreciate the positive assessment of our revised manuscript and look forward to publication.

Reviewers' Comments:

Reviewer #2:

Remarks to the Author:

All good on my side with these last little adjustments.